# Personalized Federated Learning of Probabilistic Models: A PAC-Bayesian Approach

**Mahrokh G. Boroujeni**                                    *mahrokh.ghoddousiboroujeni@epfl.ch*
*Institute of Mechanical Engineering*
*EPFL, Switzerland*

**Andreas Krause**                                                        *krausea@ethz.ch*
*Department of Computer Science*
*ETH Zürich, Switzerland*

**Giancarlo Ferrari-Trecate**                          *giancarlo.ferraritrecate@epfl.ch*
*Institute of Mechanical Engineering*
*EPFL, Switzerland*

**Reviewed on OpenReview:** *https://openreview.net/forum?id=ZMliWjMCor*

## Abstract

Federated Learning (FL) aims to infer a shared model from private and decentralized data stored by multiple clients. Personalized FL (PFL) enhances the model's fit for each client by adapting the global model to the clients. A significant level of personalization is required for highly heterogeneous clients but can be challenging to achieve, especially when clients' datasets are small. To address this issue, we introduce the PAC-PFL framework for PFL of probabilistic models. PAC-PFL infers a shared hyper-posterior and treats each client's posterior inference as the personalization step. Unlike previous PFL algorithms, PAC-PFL does not regularize all personalized models towards a single shared model, thereby greatly enhancing its personalization flexibility. By establishing and minimizing a PAC-Bayesian generalization bound on the average true loss of clients, PAC-PFL effectively mitigates overfitting even in data-poor scenarios. Additionally, PAC-PFL provides generalization bounds for new clients joining later. PAC-PFL achieves accurate and well-calibrated predictions, as supported by our experiments[1].

## 1 Introduction

*Federated Learning (FL)* enables collaborative learning across decentralized datasets stored on end-devices, known as *clients*, without requiring raw data to be shared (Konečný et al., 2015). The primary objective of FL is to train a *global* model that performs well across all clients. The training process is orchestrated by a trusted *server*, which iteratively distributes the current model to a subset of clients, aggregates their locally computed updates, and refines the model for subsequent rounds. By restricting access to only the communicated updates rather than the raw data itself, FL enhances privacy and reduces communication overhead compared to traditional centralized approaches.

A key challenge in FL is the heterogeneity of clients' datasets, which violates the i.i.d. assumption required for training a global model (Kairouz et al., 2021; Li et al., 2020a). This often leads to convergence difficulties or suboptimal performance (Li et al., 2020b). To address this issue, *Personalized FL* (PFL) introduces a *personalization* step to adapt the global model to the specific data of individual clients. This step is critical in many real-world federated datasets, as they typically involve heterogeneous clients (Wen et al., 2022).

---

[1]The codebase for our algorithm is available on `https://sites.google.com/view/pac-pfl`.

The growing impact of PFL has been highlighted in diverse applications, including image classification, regression, text analysis, and recommendation systems (Chen et al., 2024).

Despite significant advancements, several critical issues remain underexplored. First, ($c1$) most PFL approaches yield point estimates, limiting their ability to quantify epistemic uncertainty, which is essential in safety-critical applications (Guo et al., 2017; Achituve et al., 2021). Second, ($c2$) personalized models are often closely tied to the global model, making them less effective in highly heterogeneous or multimodal scenarios. Third, ($c3$) many methods suffer from performance degradation when client datasets are small. Finally, ($c4$) few approaches account for the progressive collection of new data over time. In this paper, we propose the *PAC-PFL* framework to tackle these challenges ($c1$–$c4$).

Our proposed framework leverages probabilistic models to address ($c1$) by accounting for uncertainty. In the considered setup, each client places a prior distribution over models and updates it based on its local data to derive a posterior distribution. A naive PFL approach is to collaboratively learn a shared prior distribution and treat posterior inference as the personalization step. However, this method conflicts with the Bayesian framework since the learned prior depends on each client's data (Box & Tiao, 1992), as demonstrated in Figure 1. We overcome this by leveraging PAC-Bayesian inference, which accommodates data-dependent priors (Rivasplata et al., 2020). Additionally, to address heterogeneity ($c2$), we introduce a novel inference approach: instead of relying on a single shared prior, we learn a *hyper-posterior* distribution over prior distributions from which clients sample their priors. By decoupling clients' posterior distributions from a single shared prior, our framework enhances adaptability to diverse data distributions.

Low-data scenarios ($c3$) are prone to overfitting, where models exhibit strong performance on training data but generalize poorly to unseen samples. PAC-PFL addresses this challenge by selecting a hyper-posterior that minimizes a bound on the generalization error. As shown in Section 4, this approach introduces a principled regularization mechanism for the hyper-posterior, enabling the framework to learn complex models while mitigating the risk of overfitting. Regarding ($c4$), PAC-PFL accommodates collecting new data over time. In practice, only a subset of clients may communicate with the server during any given iteration, and these clients may gather new data locally before communicating again with the server. Such data can be utilized for personalization but not for updating the shared hyper-posterior. Furthermore, PAC-PFL supports *new clients*—devices joining the system later that have never communicated with the server. A distinct generalization bound applies to this group, underscoring the importance of differentiating between existing and new clients. Table 1 summarizes the introduced challenges ($c1$–$c4$) and how they are addressed both in theory and through experimental validation.

We evaluate PAC-PFL on Gaussian Process (GP) regression and Bayesian Neural Network (BNN) classification as representative examples of probabilistic models. Our experiments demonstrate that PAC-PFL yields accurate and well-calibrated predictions ($c1$), even in highly heterogeneous ($c2$) and data-poor ($c3$) scenarios. Furthermore, we illustrate that PAC-PFL facilitates positive transfer learning from existing to new clients ($c4$). We empirically showcase the effectiveness of learning a hyper-posterior, instead of a single prior, in highly heterogeneous cases. Lastly, we provide an interpretation of our method through Jaynes' principle of maximum entropy (Jaynes, 1957) in Appendix 8.1.

## 2 Related work and novel contributions

**Meta-PFL.** PAC-PFL belongs to the meta-PFL category (Kulkarni et al., 2020). Meta-learning, developed independently of FL, involves training a global model on various related learning problems (tasks), which can be efficiently fine-tuned for a new task. The link between meta-learning and PFL was first explored in Jiang et al. (2019), where they proposed simultaneously learning the global model and its personalization. Other approaches have investigated learning a global model initialization that performs well after being personalized by individual clients (Khodak et al., 2019; Fallah et al., 2020; Chen et al., 2018). In Fallah et al. (2020), a scalable algorithm, called *MAML*, is proposed that limits the personalization step to one or a few gradient descent steps. However, when clients have small datasets, the personalized models tend to remain close to the shared initialization, resulting in a strong resemblance between them. Consequently, these methods may lack the necessary personalization capability in highly heterogeneous cases with small client datasets. Additionally, all these methods are frequentist, specific to parametric models, and lack generalization bounds.

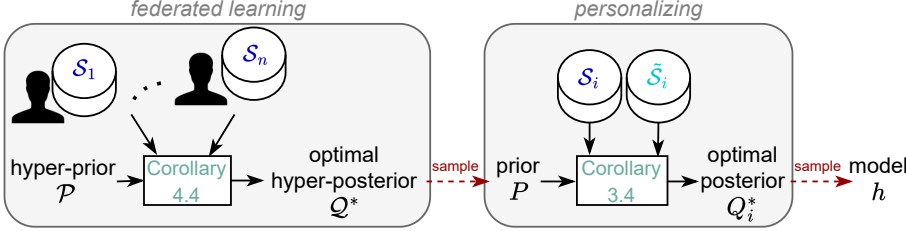

Figure 1: Illustration of the proposed PAC-PFL framework. For a given hyper-prior distribution $\mathcal{P}$, the server computes the optimal hyper-posterior $\mathcal{Q}^*$ as per Corollary 4.4 through communication with clients owning datasets $\mathcal{S}_1, \ldots, \mathcal{S}_n$. During personalization, each client $i$ draws a prior distribution $P$ from $\mathcal{Q}^*$, combining it with its local dataset $\mathcal{S}_i$ and potentially new data $\tilde{\mathcal{S}}_i$ to derive the optimal posterior distribution $Q_i^*$ according to Corollary 3.4. The client then samples a model from $Q_i^*$ for making predictions. Note that the prior $P$ depends on each client's data, $\mathcal{S}_i$, conflicting with the Bayesian framework and necessitating proper consideration, as discussed in Section 3.

Table 1: Challenges, solutions proposed by PAC-PFL, and datasets representing them. The number after each dataset name indicates the number of data samples per client. Detailed descriptions of these datasets can be found in Appendix 8.5.

| Challenge | Solution | Dataset |
|---|---|---|
| ($c1$) quantify uncertainty | probabilistic modeling | all |
| ($c2$) high heterogeneity | formulate a hyper-posterior | PV-EW (150), PV-EW (610) |
| ($c3$) overfitting | minimize a generalization bound | PV-EW (150), PV-S (150), FEMNIST (20) |
| ($c4$) new data | incorporate it in personalization | PV-EW (150), PV-EW (610), PV-S (150), PV-S (610) |

**FL for probabilistic models.** Several FL methods aim to learn a global posterior without personalization (Bui et al., 2018; Al-Shedivat et al., 2020; Kassab & Simeone, 2022). Learning personalized posteriors is studied in Corinzia & Buhmann (2019), where the posterior for each client is decomposed into global and per-client variational factors. However, this method encounters scalability challenges in systems with numerous clients, as personalized models are refined sequentially. As mentioned in Section 1, an alternative approach is to learn a shared prior and consider posterior inference as the personalization step. This idea is adopted in methods like *pFedBayes* (Zhang et al., 2022) and *pFedGP* (Achituve et al., 2021). In pFedBayes, clients compute personalized posteriors by optimizing their data likelihood while adding regularization towards the global prior. The server calculates the prior by minimizing the average client loss. Due to relying on bi-level optimization, pFedBayes imposes a significant computational burden.

The closest work to our approach is pFedGP which learns a global GP prior by maximizing the average Log Marginal Likelihood (LML) across all clients. Consequently, clients perform posterior inference for personalization. However, pFedGP only focuses on learning the covariance of the prior and sets the prior mean to zero. The reason for ignoring the mean may be that the LML implicitly regularizes the covariance, while the mean can quickly overfit (Fortuin et al., 2020). Both pFedBayes and pFedGP may exhibit limited personalization flexibility, as they utilize the same prior distribution for all clients. Additionally, their inference procedures are heuristic due to deviating from the Bayesian setup by using data-dependent priors.

**PAC-Bayesian meta-learning.** The setup described in Section 1 is related to meta-learning within the PAC-Bayesian framework (Amit & Meir, 2018; Pentina & Lampert, 2014; Rothfuss et al., 2021). In this scheme, a meta-learner is presented with a sequence of heterogeneous learning tasks, corresponding to existing clients in PFL, and aims to facilitate posterior inference for an unseen task, corresponding to a new client. The meta-learner employs the previous tasks' data to learn a hyper-posterior distribution that leads to good generalization to the new task, establishing a data flow from previous tasks to the new task.

**PAC-Bayesian federated-learning.** A PAC-Bayesian PFL algorithm introduced by Jobic et al. (2023) optimizes both the parameters of a shared prior distribution and personalized posterior distributions by minimizing non-vacuous generalization bounds from McAllester (2003). While PAC-PFL accommodates arbitrary distributions, this method assumes Gaussian prior and posterior distributions. To avoid data-dependent priors, it splits each client's dataset: one half to learn the prior and the other to learn the posterior, sacrificing data efficiency. This approach evaluates the prior based on the loss incurred when sampling models directly from it. However, an alternative strategy would be to evaluate the loss of models drawn from the posterior distribution corresponding to this prior, potentially enhancing performance by directly optimizing the effectiveness of personalized posteriors.

**Novelties** We propose PAC-PFL, a probabilistic PFL framework that addresses challenges ($c1$)-($c4$) while introducing two significant advancements over existing methods. First, PAC-PFL improves upon Bayesian and PAC-Bayesian PFL approaches by offering a theoretically rigorous and data-efficient inference pipeline that eliminates the need for data splitting. This is achieved through PAC-Bayesian methods allowing for data-dependent priors that are differentially private (Dziugaite & Roy, 2018), along with an analysis of the proposed method's privacy properties. Second, unlike PAC-Bayesian meta-learning approaches that primarily benefit new clients (tasks), PAC-PFL incentivizes existing clients to collaborate in the framework by learning a hyper-posterior distribution tailored for their data. To achieve this, we define a novel loss function that evaluates the hyper-posterior based on the accuracy of models derived via the pipeline in Figure 1 for existing clients. We then derive a generalization bound for this loss function, focusing on unseen data from existing clients rather than new clients, as in meta-learning. With these advancements, PAC-PFL provides a principled solution for addressing existing challenges in PFL.

## 3 Preliminaries and notation

We start by presenting the notation and key concepts. A summary of the introduced notation is provided in Table 4. Consider a set of $n \in \mathbb{N}$ clients, referred to as *existing clients*, each observing samples of feature-target pairs $(\mathbf{x}, y)$, where $\mathbf{x} \in R_{\mathbf{x}} \subset \mathbb{R}^d$ represents the features and $y \in R_y$ denotes the target values. The feature space $R_{\mathbf{x}}$ and the target space $R_y$ are identical across clients, and for simplicity, we assume the target $y$ is scalar, i.e., $R_y \subset \mathbb{R}$. Each client has access to two distinct i.i.d. datasets: $\mathcal{S}_i \sim \mathcal{D}_i^{m_i}$, containing $m_i \in \mathbb{N}$ samples, and $\tilde{\mathcal{S}}_i \sim \mathcal{D}_i^{\tilde{m}_i}$, containing $\tilde{m}_i \in \mathbb{N} \cup \{0\}$ samples. The first dataset, $\mathcal{S}_i$, is non-empty and is used by client $i$ during both FL and personalization phases. The second dataset, $\tilde{\mathcal{S}}_i$, consists of samples acquired later, which are used exclusively for the personalization phase and may be empty. We assume that $\tilde{m}_i \le m_i$ for all clients $i$, and we denote the number of clients for which $\tilde{m}_i > 0$ as $n_2 \in \{0, \ldots, n\}$.

The data held by each client $i$ is sampled from an unknown data distribution $\mathcal{D}_i$ over the support $R_{\mathbf{z}} = R_{\mathbf{x}} \times R_y$. To account for client heterogeneity, we allow both the distributions and the number of samples to vary across clients, i.e., $\mathcal{D}_i \neq \mathcal{D}_j$ and $m_i \neq m_j$. We capture the relatedness among clients through a distribution $\mathcal{T}$ over the data distributions and sample sizes, such that $(\mathcal{D}_i, m_i) \sim \mathcal{T}$. For convenience, we introduce the following notations: $\mathcal{S} \coloneqq \{\mathcal{S}_i\}_{i=1}^n$, $\mathcal{D} \coloneqq \{\mathcal{D}_i\}_{i=1}^n$, $\mathbf{m} \coloneqq [m_1, \cdots, m_n]$, and $\tilde{\mathbf{m}} \coloneqq [\tilde{m}_1, \cdots, \tilde{m}_n]$.

Each client $i$ aims to learn a hypothesis function $h_i : R_{\mathbf{x}} \to R_y$ for predicting the label $y_*$ of an unseen input $\mathbf{x}_*$. The error incurred by $h_i$ at $(\mathbf{x}_*, y_*)$ is measured by a loss function which we assume to be bounded, $\ell : \mathcal{H} \times R_{\mathbf{z}} \to [a, b]$. Since the data distribution, $\mathcal{D}_i$, is unknown, standard methods select the *best* hypothesis according to the set of observed samples, $\mathcal{S}_i \cup \tilde{\mathcal{S}}_i$; for instance, by minimizing the empirical risk associated with $\ell$.

Using a single hypothesis based on limited observations leads to epistemic uncertainty (Draper, 1995) and overconfident target estimation (Kass & Raftery, 1995). To address this issue, the *PAC-Bayesian* framework (McAllester, 1999) places probability distributions over the hypothesis space, $\mathcal{H}$, and combines all possible hypotheses sampled from these distributions for making inferences. Two such distributions are enlisted: a *prior* distribution $P$, and a *posterior* distribution $Q_i$ which depends on the prior and the observations through a mapping, $Q_i \coloneqq \mathbb{Q}(P, \mathcal{S}_i \cup \tilde{\mathcal{S}}_i)$. While the prior and posterior terms resemble the Bayesian terminology, the posterior mapping, $\mathbb{Q}$, is not (necessarily) obtained through Bayes' theorem. Additionally, recent advances in PAC-Bayesian analysis enable employing priors that *slightly* depend on the data to make the learning pipeline more data-driven, which is contrary to the Bayesian formalism (Rivasplata et al., 2020).

**Client-level PAC-Bayesian bounds.** The flexibility in choosing the mapping $\mathbb{Q}$ can be exploited to achieve desired characteristics. Ideally, clients aim to choose a $\mathbb{Q}$ that minimizes the *true risk*,

$$\mathcal{L}^C\big(\mathbb{Q}(P, \mathcal{S}_i \cup \tilde{\mathcal{S}}_i), \mathcal{D}_i\big) \coloneqq \mathbb{E}_{h \sim \mathbb{Q}(P, \mathcal{S}_i \cup \tilde{\mathcal{S}}_i)} \mathbb{E}_{\mathbf{z} \sim \mathcal{D}_i}\big[\ell(h, \mathbf{z})\big], \tag{1}$$

where superscript $C$ indicates that (1) is calculated by a client. In most cases, $\mathcal{D}_i$ is unknown and the true loss is approximated by its empirical counterpart,

$$\hat{\mathcal{L}}^C\big(\mathbb{Q}(P, \mathcal{S}_i \cup \tilde{\mathcal{S}}_i), \mathcal{S}_i \cup \tilde{\mathcal{S}}_i\big) \coloneqq \mathbb{E}_{h \sim \mathbb{Q}(P, \mathcal{S}_i \cup \tilde{\mathcal{S}}_i)}\Big[\frac{1}{m_i + \tilde{m}_i} \sum_{\mathbf{z} \in \mathcal{S}_i \cup \tilde{\mathcal{S}}_i} \ell(h, \mathbf{z})\Big]. \tag{2}$$

PAC-Bayes theory (McAllester, 1999) provides a guarantee on the worst loss clients may suffer by upper bounding the unknown true risk in (1) based on the empirical risk in (2). The original PAC bound by McAllester (1999) assumes that the prior, $P$, is independent of the data. However, in our setup, the prior is obtained from the FL algorithm acting on the data subset $\mathcal{S}_i$, hence, is *data-dependent*. Dziugaite & Roy (2018) propose a recipe for adapting any PAC bound with data-free prior to a data-dependent prior that is stable to slight changes in the data, formalized by the notion of *Differential Privacy (DP)* defined below.

**Definition 3.1** ((Dwork & Roth, 2014)). Let $\epsilon_i \in \mathbb{R}_+$ and $\mathcal{A}$ be a *randomized* algorithm that generates a stochastic output given an input dataset. The algorithm $\mathcal{A}$ preserves $\epsilon_i$-DP if for all datasets $\mathcal{S}_i$ and $\hat{\mathcal{S}}_i$ that differ in a single data sample and all subsets $\mathcal{O}$ of possible outcomes of $\mathcal{A}$,

$$e^{-\epsilon_i} \leq \frac{Pr[\mathcal{A}(\hat{\mathcal{S}}_i) \in \mathcal{O}]}{Pr[\mathcal{A}(\mathcal{S}_i) \in \mathcal{O}]\big|} \leq e^{\epsilon_i}, \tag{3}$$

where the probability is w.r.t the algorithm's randomness.

Intuitively, (3) bounds the stability of $\mathcal{A}$ to changing a single sample of $\mathcal{S}$ (Dwork et al., 2015). The privacy/stability level is controlled by $\epsilon$: for a smaller $\epsilon$, $\mathcal{A}$ is more private/stable.

To state a PAC bound for client $i$, we regard the federated pipeline as a randomized algorithm that takes $\mathcal{S}_i$ as input and outputs $P = \mathcal{A}_{\mathcal{S} \setminus \mathcal{S}_i}(\mathcal{S}_i)$. The subscript $\mathcal{S} \setminus \mathcal{S}_i$ contains samples from all clients except $i$ and emphasizes that $P$ depends on data from other clients likewise. However, DP is only studied for the argument, $\mathcal{S}_i$. Randomization in $\mathcal{A}_{\mathcal{S} \setminus \mathcal{S}_i}$ arises from sampling the prior from the hyper-posterior.

Assume the prior is obtained by $\mathcal{A}$ preserving $\epsilon$-DP. We apply the method of Dziugaite & Roy (2018) on a bound due to Alquier et al. (2016) to derive the following theorem.

**Theorem 3.2.** *Fix a data-dependent prior $P$ obtained by an $\epsilon_i$-DP algorithm, a data distribution $\mathcal{D}_i$, and a bounded loss function $\ell(\cdot, \cdot) \in [a, b]$. For every $\beta > 0$, confidence level $\delta \in (0, 1]$, and posterior $Q_i = \mathbb{Q}(P, \mathcal{S}_i \cup \tilde{\mathcal{S}}_i)$, the inequality*

$$\mathcal{L}^C\big(Q_i, \mathcal{D}_i\big) \leq \hat{\mathcal{L}}^C\big(Q_i, \mathcal{S}_i \cup \tilde{\mathcal{S}}_i\big) + \frac{1}{\beta}\Big(KL(Q_i \| P) + \frac{\beta^2(b-a)^2}{8(m_i + \tilde{m}_i)} + I(\epsilon_i, m_i, \delta) + \ln\big(\frac{1}{\delta}\big)\Big) \tag{4}$$

*holds with probability at least $1 - \delta$ over $\mathcal{S}_i \sim \mathcal{D}_i^{m_i}$ and $\tilde{\mathcal{S}}_i \sim \mathcal{D}_i^{\tilde{m}_i}$. In the above, $I(\epsilon_i, m_i, \delta) = 0.5 m_i \epsilon_i^2 + \epsilon_i \sqrt{0.5 m_i \ln(4/\delta)} + \ln(2)$, and does not depend on the posterior. It is assumed that the KL divergence between $Q_i$ and $P$ exists and is denoted by $KL(Q_i \| P)$.*

*Remark* 3.3. The term $I(\epsilon_i, m_i, \delta)$ is the only difference with the bound of Alquier et al. (2016) with a data-free prior.

We refer to Theorem 3.2 as the *client-level bound*.

**Optimal posterior.** The bound in (4) holds for all $Q_i$ and thus can be minimized w.r.t $Q_i$ to obtain the tightest upper bound. Since $\epsilon_i$ only reflects through the term $I$ in (4), the optimal posterior is the same as the minimizer of the bound with a data-free prior derived in Catoni (2007).

**Corollary 3.4** ((Catoni, 2007))**.** *Given a prior, $P$, obtained through an $\epsilon_i$-DP algorithm and observations, $\mathcal{S}_i \cup \tilde{\mathcal{S}}_i$, the optimal posterior minimizing the right-hand side of (4) is the Gibbs distribution:*

$$Q_i^*(h) = \frac{P(h)\, e^{(\frac{-\beta}{m_i + \tilde{m}_i} \sum_{\mathbf{z} \in \mathcal{S}_i \cup \tilde{\mathcal{S}}_i} \ell(h, \mathbf{z}))}}{Z_\beta^C(P, \mathcal{S}_i \cup \tilde{\mathcal{S}}_i)}, \tag{5}$$

$$Z_\beta^C(P, \mathcal{S}_i \cup \tilde{\mathcal{S}}_i) := \mathbb{E}_{h \sim P}\, e^{(\frac{-\beta}{m_i + \tilde{m}_i} \sum_{\mathbf{z} \in \mathcal{S}_i \cup \tilde{\mathcal{S}}_i} \ell(h, \mathbf{z}))},$$

*where $Z_\beta^C(P, \mathcal{S}_i \cup \tilde{\mathcal{S}}_i)$ is a normalization constant. We denote the dependence of the optimal posterior on the prior and data using the operator $\mathbb{Q}^*$: $Q_i^* = \mathbb{Q}^*(P, \mathcal{S}_i \cup \tilde{\mathcal{S}}_i)$.*

With $\beta = m_i + \tilde{m}_i$ and the negative log-likelihood loss, $\ell(h, \mathbf{z}) = -\ln \Pr[\mathbf{z}|h]$, $Z_\beta^C$ and $Q_i^*$ simplify to the LML and the Bayes posterior respectively (Guedj, 2019).

Plugging in the closed-form formula of the optimal posterior (5) into the client-level bound obtains:

$$\mathcal{L}^C\big(\mathbb{Q}^*(P, \mathcal{S}_i \cup \tilde{\mathcal{S}}_i), \mathcal{D}_i\big) \leq \frac{1}{\beta}\Big( -\ln Z_\beta(P, \mathcal{S}_i \cup \tilde{\mathcal{S}}_i) + \frac{\beta^2 (b-a)^2}{8(m_i + \tilde{m}_i)} + I(\epsilon_i, m_i, \delta) + \ln\big(\frac{1}{\delta}\big)\Big), \tag{6}$$

holding with probability at least $1 - \delta$ over $\mathcal{S}_i \sim \mathcal{D}_i^{m_i}, \tilde{\mathcal{S}}_i \sim \mathcal{D}_i^{\tilde{m}_i}$. The simplified bound (6) removes the explicit dependence on $Q_i$ and is tighter than the generic bound per (4).

In the rest of this paper, we assume that clients utilize $Q_i^*$ whenever the privacy requirement of Theorem 3.2 is satisfied. To highlight the generality of our approach, we note that Bayesian inference is a subcase of the assumed setup.

## 4 Theoretical framework for PAC-Bayesian federated learning

In this section, we present a data-driven approach for obtaining a prior distribution. Rather than focusing on a single prior, we propose learning distributions over priors, enabling a more flexible and robust framework. To formalize this, we introduce the following key concepts:

**Definition 4.1** (Hyper-distributions)**.** A *hyper-prior*, $\mathcal{P}$, is a distribution over prior distributions that is independent of clients' datasets. Conversely, a *hyper-posterior*, $\mathcal{Q}$, is a distribution over priors that can rely on the data of existing clients, $\mathcal{S}_1, \cdots, \mathcal{S}_n$.

A trusted server communicates with the existing clients to extract common knowledge in the form of a hyper-posterior distribution without directly accessing their datasets (as enforced by FL). At each communication round, the server sends the hyper-posterior to the clients. The clients proceed by repeatedly drawing a prior from the received hyper-posterior and using it to calculate the optimal posterior per (5), involving any potential additional samples, $\tilde{\mathcal{S}}_i$ in the inference procedure. The goal is to find the *optimal hyper-posterior*, $\mathcal{Q}^*$, such that the posterior obtained through the described pipeline has an average low true risk (1) for all clients. An overview of the setup is depicted in Figure 1.

In the sequel, we consider hyper-posteriors, $\mathcal{Q}$, that satisfy the condition that for a finite $\epsilon \in \mathbb{R}_+$, sampling $P$ from $\mathcal{Q}$ preserves $\epsilon$-DP for all clients. This assumption enables us to use Corollary 3.4 and is rather weak as $\epsilon$ can be arbitrarily, albeit not infinitely, large. Analogous to the procedure in Section 3, we establish PAC bounds for the pair $\mathcal{P}$ and $\mathcal{Q}$. Next, we derive the closed-form formula for the optimal hyper-posterior, $\mathcal{Q}^*$, which minimizes the PAC bound, and verify that $\mathcal{Q}^*$ satisfies the privacy assumption outlined above. Finally, we establish a PAC bound for new clients who sample their priors from $\mathcal{Q}^*$ without participating in the federated learning process. An interpretation through the principle of maximum entropy (Jaynes, 1957) and all proofs are provided in Appendices 8.1 and 8.2, respectively.

**Server-level PAC-Bayesian bound.** Following the introduced inference setup, we evaluate the quality of a hyper-posterior distribution $\mathcal{Q}$ using the *server-level true risk*, defined as:

$$\mathcal{L}^S(\mathcal{Q}, \mathcal{D}, \mathcal{S}, \tilde{\mathbf{m}}) := \frac{1}{n} \sum_{i=1}^n \mathbb{E}_{P \sim \mathcal{Q}} \mathbb{E}_{\tilde{\mathcal{S}}_i \sim \mathcal{D}_i^{\tilde{m}_i}} \mathcal{L}^C(\mathbb{Q}^*(P, \mathcal{S}_i \cup \tilde{\mathcal{S}}_i), \mathcal{D}_i). \tag{7}$$

This metric computes the average true loss across all clients, considering all possible sets of additional samples, $\tilde{\mathcal{S}}$, of size $\tilde{\mathbf{m}}$. However, the server-level true risk is intractable due to its reliance on expectations over the underlying data distributions $\mathcal{D}$. To address this, we approximate it with an empirical estimate:

$$\hat{\mathcal{L}}^S(\mathcal{Q}, \mathcal{S}) := \frac{1}{n} \sum_{i=1}^n \mathbb{E}_{P \sim \mathcal{Q}} \hat{\mathcal{L}}^C(\mathbb{Q}^*(P, \mathcal{S}_i), \mathcal{S}_i), \tag{8}$$

where observed samples replace the unobserved future data. We refer to (8) as the *server-level empirical loss*.

Below, we present our first main contribution, which is a PAC bound on server-level loss.

**Theorem 4.2.** *Let $\ell(\cdot, \cdot) \in [a, b]$ be a bounded loss. Define:*

$$\Delta_i := \frac{1}{n} \min \left\{ b - a, b \left( e^{\frac{2\beta \tilde{m}_i}{m_i + \tilde{m}_i}(b-a)} - e^{\frac{-2\beta \tilde{m}_i}{m_i + \tilde{m}_i}(b-a)} \right) \right\},$$

*for all $i \in \{1, \cdots, n\}$. Assume clients employ the optimal posterior with parameter $\beta \geq 1/n$. Let $\mathcal{Q}$ be a hyper-posterior such that sampling $P$ from $\mathcal{Q}$ preserves $\epsilon$-DP for all clients. For every hyper-prior $\mathcal{P}$ independent from $\mathcal{S}$, $\upsilon > 0$, $\lambda > n_2 + \upsilon$, and confidence level $\delta \in (0, 1)$,*

$$\mathcal{L}^S(\mathcal{Q}, \mathcal{D}, \mathcal{S}, \tilde{\mathbf{m}}) \leq \frac{-1}{n\beta} \sum_{i=1}^n \mathbb{E}_{P \sim \mathcal{Q}} \ln Z_\beta^C(P, \mathcal{S}_i) + \left( \frac{1}{n\beta} + \frac{n_2 + \upsilon}{\lambda} \right) KL(\mathcal{Q} \| \mathcal{P})$$

$$+ \frac{\beta(b-a)^2}{8n} \sum_{i=1}^n \frac{1}{m_i} + \frac{\lambda \sum_{i=1}^n \Delta_i^2}{8(n_2 + \upsilon)} + \frac{1}{\sqrt{n}} \ln \left( \frac{1}{\delta} \right), \tag{9}$$

*holds with probability at least $1 - \delta$ over $\mathcal{S}_i \sim \mathcal{D}_i^{m_i}$ and $\tilde{\mathcal{S}}_i \sim \mathcal{D}_i^{\tilde{m}_i}$, for $i = 1, \cdots, n$. The constant $\upsilon$ is chosen to be very small and avoids numerical issues when $n_2 = 0$.*

The first term on the right-hand side of (9) falls within $[a, b]$ due to our bounded loss assumption. The summand $-1/\beta \ln Z_\beta^C(P, \mathcal{S}_i)$ in this expression decreases when the prior $P$ "aligns better" with the dataset $\mathcal{S}_i$, favoring hypotheses with lower empirical costs over $\mathcal{S}_i$. Consequently, the first term on the right-hand side decreases if $\mathcal{Q}$ assigns higher probability to priors that are, on average, "better aligned" with all clients' data. The KL-divergence term in (9) regularizes the hyper-posterior towards the hyper-prior and avoids overfitting to the existing clients when $n$ is small. The bound depends on the number of clients and the available dataset sizes, $n$ and $\mathbf{m}$. The bound in (9) tightens with increasing the number of samples involved in calculating the empirical loss, $m_i$, demonstrating consistency with the PAC-Bayes framework. Finally, the bound relies on predictions for the number of clients with new samples and the corresponding new sample sizes, $n_2$ and $\tilde{\mathbf{m}}$. While the server knows $n$ and $\mathbf{m}$, the estimates for $n_2$ and $\tilde{\mathbf{m}}$ might be coarse. The next lemma states that a pessimistic forecast leads to a looser upper bound.

**Lemma 4.3.** *If the number of new samples of client $i$, $\tilde{m}_i \geq 0$, is unknown, Theorem 4.2 holds when replacing $\Delta_i$ with $(b - a)/n$ and counting client $i$ in $n_2$, i.e., as if $\tilde{m}_i > 0$.*

The asymptotic behavior and non-vacuousness of the client and server-level bounds are addressed in Appendix 8.3.1.

**Optimal hyper-posterior.** Our algorithm picks the *optimal hyper-posterior*, $\mathcal{Q}^*$, leading to the lowest upper bound on the server-level true risk per (4). Inspecting the structural similarity between the server and client-level bounds in (4) and (9), we arrive at a closed-form formula for $\mathcal{Q}^*$.

**Corollary 4.4.** *When clients use the optimal posterior, $Q_i^*$, the optimal hyper-posterior is a Gibbs distribution with parameter $\tau = \lambda/\big(\lambda + \beta n(n_2 + \upsilon)\big)$:*

$$\mathcal{Q}^*(P) = \mathcal{P}(P) \cdot \exp\Big(\tau \sum_{i=1}^{n} \ln\big(Z_\beta^C(P, \mathcal{S}_i)\big)\Big)/Z_\tau^S(\mathcal{P}, \mathcal{S}),$$

*where $Z_\tau^S(\mathcal{P}, \mathcal{S}) \coloneqq \mathbb{E}_{P\sim\mathcal{P}} \exp\big(\tau \sum_{i=1}^{n} \ln\big(Z_\beta^C(P, \mathcal{S}_i)\big)\big)$ is a normalization constant.*

The parameter $\tau$ depends on the number of clients, $n$ and $n_2$, but not on the number of samples, $\mathbf{m}$ and $\tilde{\mathbf{m}}$. If $n_2$ is unknown, it can be replaced consistently with Lemma 4.3. In this case, a looser upper bound would be minimized.

The privacy of sampling a prior from $\mathcal{Q}^*$ is crucial to obtain $\epsilon_i$ for plugging it into the client-level bound (6) and for employing Theorem 4.2. We rely on a result by Mir (2012) that proves the DP of sampling from the Gibbs distribution.

**Lemma 4.5.** *A prior sampled from $\mathcal{Q}^*$ preserves $\epsilon_i$-DP for client $i$, where $\epsilon_i = 2\beta\tau(b-a)/m_i$ .*

As a result, $\mathcal{Q}^*$ satisfies the privacy assumption of Theorem 4.2 with $\epsilon = \max_{i\in\{1,\cdots,n\}} \epsilon_i$. Additional insights into the role of DP in our framework are provided in Appendix 8.4.

**PAC-Bayesian bound for new clients.** So far, we considered a fixed set of *existing* clients who participate in training the optimal hyper-posterior. In a realistic FL setup, there might be *new* clients (see c4) who join the system later and hence, do not engage in federated training. A new client holds a presumably small set of samples which leads to overfitting. Assuming the existing and new clients are similar, it is constructive for the new clients to readily employ $\mathcal{Q}^*$ without having contributed to training it. In this section, we establish a PAC bound for such new clients.

Section 3 introduced the distribution $\mathcal{T}$ to capture the similarity among existing clients. Consistently, we expect that a new client $\iota$ is sampled from the same distribution, $(\mathcal{D}_\iota, \tilde{m}_\iota) \sim \mathcal{T}$. In line with previous notation, $\tilde{\mathcal{S}}_\iota \sim \mathcal{D}_\iota^{\tilde{m}_\iota}$ is a dataset of size $\tilde{m}_\iota$ employed by client $\iota$ for personalization but excluded from FL. Our second PAC-Bayesian bound applies to new clients and is presented below.

**Lemma 4.6.** *For a new client $\iota$ sampled from $\mathcal{T}$ adopting $\mathbb{Q}^*$ and $\mathcal{Q}^*$ as per Corollaries 3.4, 4.4, it holds with probability at least $1 - \delta$ over $(\mathcal{D}_\iota, \tilde{m}_\iota) \sim \mathcal{T}$ and $\tilde{\mathcal{S}}_\iota \sim \mathcal{D}_\iota^{\tilde{m}_\iota}$ that:*

$$\mathbb{E}_{(\mathcal{D}_\iota, \tilde{m}_\iota)\sim\mathcal{T}} \mathbb{E}_{\tilde{\mathcal{S}}_\iota \sim \mathcal{D}_\iota^{\tilde{m}_\iota}} \mathbb{E}_{P\sim\mathcal{Q}^*} \mathcal{L}^C\big(\mathbb{Q}^*(P, \tilde{\mathcal{S}}_\iota), \mathcal{D}_\iota\big) \leq -\big(\frac{1}{n\beta} + \frac{n_2 + \upsilon}{\lambda}\big) \ln Z_\tau^S(\mathcal{P}, \mathcal{S})$$

$$+ \frac{(b-a)^2}{8n}\big(\beta \sum_{i=1}^{n} \frac{1}{m_i} + \frac{\lambda}{n_2 + \upsilon}\big) + \frac{1}{\sqrt{n}} \ln(\frac{1}{\delta}).$$

Since $\mathcal{Q}^*$ is tailored to minimize the server-level bound for existing clients, the bound in Lemma 4.6 is looser than that of Theorem 4.2 with $\mathcal{Q} = \mathcal{Q}^*$ (proof in Appendix 8.2.5). This motivates the clients to actively engage in training $\mathcal{Q}^*$ rather than readily employing the learned hyper-posterior.

## 5 Practical federated implementation

Section 3 justified learning a distribution over priors for heterogeneous clients, recognizing that selecting a single best prior may not be accurate or feasible. Consequently, Corollary 4.4 provided the optimal hyper-posterior. This section tackles computational constraints at both the client and server levels and introduces a practical PFL algorithm.

**Calculating the LML at the client level.** The formula for $\mathcal{Q}^*$ in Corollary 4.4 relies on $Z_\beta^C(P, \mathcal{S}_i)$ for $i \in \{0, \cdots, n\}$. As per Corollary 3.4, calculating $Z_\beta^C(P, \mathcal{S}_i)$ entails computing the expectation over all hypotheses sampled from the prior, which is generally intractable. For the negative log-likelihood loss, we set $\beta = m_i + \tilde{m}_i$ , making $Z_\beta$ align with the LML, as discussed in Section 3. We calculate the LML for two

---

**Algorithm 1** PAC-PFL executed by the server

---

1: **Input:** number of SVGD priors $k$, hyper-prior $\mathcal{P}$, parameter $\tau$, number of iterations $T$, number of clients per iteration $c$, mini-batch size $b$, learning rate $\eta$

2: Initialize priors $P_{\phi_1}, \ldots, P_{\phi_k} \overset{i.i.d}{\sim} \mathcal{P}^k$           $\triangleright$ *Initialize*

3: **for** $t = 1$ to $T$ **do**

4:      Select a random subset $\mathcal{C}_t$ of $c$ clients

5:      **for** each selected client $i$ in $\mathcal{C}_t$ **in parallel do**

6:          $\boldsymbol{G}_i \leftarrow Client\_Update(b, \boldsymbol{\phi}_1, \ldots, \boldsymbol{\phi}_k)$       $\triangleright$ *Collect client updates*

7:      $\boldsymbol{G} \leftarrow \frac{1}{c} \sum_{i \in \mathcal{C}_t} \boldsymbol{G}_i$         $\triangleright$ *Aggregate client updates*

8:      **for** $\kappa = 1$ to $k$ **do**

9:          $\nabla_{\boldsymbol{\phi}_\kappa} \ln \mathcal{Q}^*(\boldsymbol{\phi}_\kappa) \leftarrow \nabla_{\boldsymbol{\phi}_\kappa} \ln \mathcal{P}(\boldsymbol{\phi}_\kappa) + \tau \, G_{\kappa *}^T$       $\triangleright$ $G_{\kappa *}$ *is the $\kappa$-th row of* $\boldsymbol{G}$

10:      **for** $\kappa = 1$ to $k$ **do**

11:          $\boldsymbol{\phi}_\kappa \leftarrow \boldsymbol{\phi}_\kappa + \frac{\eta}{k} \sum_{l=1}^{k} \left( k_{SVGD}(\boldsymbol{\phi}_l, \boldsymbol{\phi}_\kappa) \nabla_{\boldsymbol{\phi}_l} \ln \mathcal{Q}^*(\boldsymbol{\phi}_l) + \nabla_{\boldsymbol{\phi}_l} k_{SVGD}(\boldsymbol{\phi}_l, \boldsymbol{\phi}_\kappa) \right)$    $\triangleright$ *SVGD update*

12: **return** $P_{\boldsymbol{\phi}_1}, \ldots, P_{\boldsymbol{\phi}_k}$        $\triangleright$ *SVGD approximation of* $\mathcal{Q}^*$

---

**Algorithm 2** *Client\_Update* for client $i$ with dataset $\mathcal{S}_i$

---

1: **Input:** mini-batch size $b$, current particles $\boldsymbol{\phi}_1, \cdots, \boldsymbol{\phi}_k$

2: Sample mini-batch $\mathcal{S}_i^{(b)}$ of size $b$ from $\mathcal{S}_i$        $\triangleright$ *Data subsampling*

3: **for** $\kappa = 1$ to $k$ **do**

4:      Compute $\nabla_{\boldsymbol{\phi}_\kappa} \ln Z_{m_i}^C(P_{\boldsymbol{\phi}_\kappa}, \mathcal{S}_i^{(b)})$ through automatic differentiation of the LML given by (33) and (38)

5: $\boldsymbol{G}_i \leftarrow [\nabla_{\boldsymbol{\phi}_1} \ln Z_{m_i}^C(P_{\boldsymbol{\phi}_1}, \mathcal{S}_i^{(b)}), \cdots, \nabla_{\boldsymbol{\phi}_k} \ln Z_{m_i}^C(P_{\boldsymbol{\phi}_k}, \mathcal{S}_i^{(b)})]^T$

6: **return** $\boldsymbol{G}_i$        $\triangleright$ *Update from client $i$*

---

scenarios: clients using GPs or BNNs. The LML is available in closed form for GPs but is intractable for BNNs. To address this, we use the approximation method described by Rothfuss et al. (2021). The formulas for calculating the LML are provided in Appendices 8.3.2 and 8.3.3.

**SVGD at the server level.** Given $Z_\beta^C(P, \mathcal{S}_i)$, $\mathcal{Q}^*$ is computable up to the constant $Z_\tau^S(\mathcal{P}, \mathcal{S})$, which leaves sampling from $\mathcal{Q}^*$ intractable. Following Rothfuss et al. (2021), we use Stein Variational Gradient Descent (SVGD) (Liu & Wang, 2016) that approximates $\mathcal{Q}^*$ as a set of particles, $P_{\boldsymbol{\phi}_1}, \cdots, P_{\boldsymbol{\phi}_k}$. Each particle $P_{\boldsymbol{\phi}_\kappa}$ is a prior parameterized by $\boldsymbol{\phi}_\kappa$. SVGD is initialized with a set of priors and then iteratively transports them to match $\mathcal{Q}^*$. This is achieved through a form of functional gradient descent on the SVGD loss (see Appendix 8.3.4), making it suitable for being integrated into an FL scheme.

As SVGD is deterministic (Liu, 2017), the inherent privacy of $\mathcal{Q}^*$ established in Lemma 4.5 is compromised. To reintroduce privacy, a conventional approach involves injecting noise into the SVGD gradients (Geyer et al., 2017). We propose a privacy-preserving variant of PAC-PFL in Appendix 8.4.

**Federated algorithm.** The pseudocode of our algorithm is presented in Algorithm 1. Initially, the server samples $P_{\boldsymbol{\phi}_1}, \cdots, P_{\boldsymbol{\phi}_k}$ from $\mathcal{P}$, defined as a zero-mean multivariate Gaussian distribution with a diagonal covariance matrix. At each iteration, the server randomly selects a subset of existing clients and sends them $\boldsymbol{\phi}_1, \cdots, \boldsymbol{\phi}_k$. The selected clients perform the *Client\_Update* sub-routine to compute the gradient of the LML with respect to the particles on a mini-batch of size $b$ of their data and send it back to the server. The server updates the particles based on the aggregated gradients, $\boldsymbol{G}$, and the learning rate, $\eta$. The particle update formula also depends on the SVGD kernel, $k_{SVGD}$, as discussed in Appendix 8.3.4. For a comprehensive list of parameters used in our theoretical results and algorithm, along with selection guidelines, please refer to Appendix 8.3.5.

Table 2: Summary of the employed datasets. The number of samples is the dataset size per client.

| Dataset | Num samples ($m$) | Num clients ($n$) | Task |
|---------|-------------------|-------------------|------|
| PV-EW (150 / 610) | 150 / 610 | 24 | regression |
| PV-S (150 / 610) | 150 / 610 | 24 | regression |
| FEMNIST (20 / 500) | 20 / $\sim 500$ | 40 | 10-way classification |
| EMNIST | $\in [516, 1954]$ | 80 | 62-way classification |
| Polynomial | 10 | 24 | regression |

## 6 Experiments

We evaluate PAC-PFL on four datasets: photovoltaic (PV) panels and Polynomial datasets for regression, alongside FEMNIST (Caldas et al., 2019) and EMNIST (Cohen et al., 2017) datasets for classification. Our algorithm consistently outperforms federated and data-centric baselines, improving the prediction accuracy and the calibration of uncertainty estimates simultaneously. These enhancements are evident in reducing the variance and mean of these metrics across existing and new clients. Our experiments demonstrate the effectiveness of the solutions proposed in Table 1 for mitigating the identified challenges.

**Datasets.** The PV dataset comprises PV generation time-series data from multiple houses within a city, with each house treated as a client. The clients exhibit heterogeneity due to variations in location, shadows, and orientation relative to the sun ($c2$). We explore two scenarios: PV-EW, featuring a bimodal distribution over clients, where half are oriented almost eastward and half almost westward, and PV-S, with all clients oriented almost southward. In both scenarios, we consider 24 existing clients and 24 new clients ($c4$) and examine $m_i = 150$ or $m_i = 610$ training samples per client. We specify $m_i$ in front of the dataset name, such as PV-EW (150). The case with $m_i = 150$ imposes ($c3$). More information about the PV dataset and the description of the Polynomial dataset are available in Appendix 8.5.

The FEMNIST dataset consists of handwritten characters from various writers, treated as clients, and we employ it for 10-way digit classification. Heterogeneity arises from distinct handwriting ($c2$). We select 40 clients and examine two scenarios: FEMNIST (20) with 20 and FEMNIST (500) with an average of 500 samples per client. In the low-data case ($c3$), we demonstrate PAC-PFL's superior performance over all baselines. In the full-data case, we highlight the scalability of our algorithm with large datasets. The EMNIST dataset is detailed in Appendix 8.5. Table 2 provides a summary of the employed datasets.

**Baselines.** We examine two probabilistic PFL methods, pFedGP (Achituve et al., 2021) and pFedBayes (Zhang et al., 2022), and two frequentist PFL methods, MAML (Fallah et al., 2020) and MTL (Evgeniou & Pontil, 2004). Additionally, we consider two non-federated approaches: Vanilla, where each client trains a model individually, and Pooled, where a single model is trained in a data-centric manner on a pooled dataset comprising all clients' data. The Pooled approach is expected to perform poorly for heterogeneous clients due to the lack of personalization. Hyper-parameters for each method are tuned using cross-validation. Further baseline details can be found in Appendix 8.6.

**Model configuration** For the PV experiment, we train a Neural Network (NN) using the MAML and MTL methods, and a GP using the PAC-PFL, pFedGP, Vanilla, and Pooled methods. Inspired by Fortuin et al. (2020); Rothfuss et al. (2021), we parameterize the GP mean and kernel with two deep NNs and consider a Gaussian likelihood. This model enhances the expressive power and scalability of GPs to high-dimensional data (Wilson et al., 2016). Further details are provided in Appendix 8.3.2.

For all classification experiments, we utilize a Bayesian Convolutional Neural Network (BCNN) (Gal & Ghahramani, 2016) for the Bayesian approaches and a Convolutional Neural Network (CNN) for the frequentist methods. All BCNNs and CNNs share the same architecture proposed by Zhang et al. (2023). Specifically, the network consists of two convolutional layers with $5 \times 5$ kernels, ReLU activation functions,

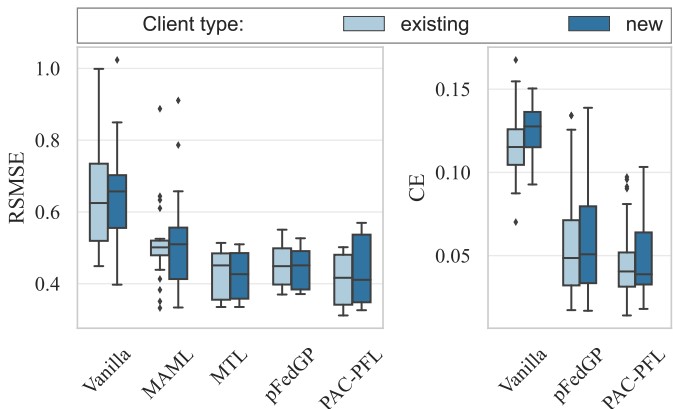

Figure 2: Box plots of test RSMSE and CE for existing and new clients in the PV-EW (150) dataset. The line within each box is the median. PAC-PFL excels in CE median, CE spread, and RSMSE median over baselines. RSMSE spread is comparable to MTL and pFedGP. Pooled GP results are not plotted due to poor performance but are reported in Appendix 8.6.

max pooling, and 10 and 62 output channels, respectively. These convolutional layers are followed by two dense layers with ReLU and SoftMax activation functions.

**Metrics.** We assess prediction accuracy and calibration for each client. For regression, we use *root standardized mean squared error* (RSMSE) that normalizes RMSE by the standard deviation of targets. For classification, we measure the percentage of correctly classified samples. Additionally, we compute the *calibration error* (CE), which quantifies the deviation of predicted confidence intervals from actual proportions of test data within those intervals (Kuleshov et al., 2018), using the formula in Rothfuss et al. (2021) (see 8.6.1). CE applies exclusively to probabilistic models and is irrelevant to frequentist baselines, such as MAML and MTL. In evaluating FL methods, the sample mean of a metric is often a biased estimate due to correlations across clients. We employ box plots to analyze the distribution of a metric across clients, offering a more comprehensive understanding.

**Results.** Figure 2 presents the results for PV-EW (150), highlighting several observations. The limited sample size adversely impacts the local Vanilla GP ($c3$). MAML does not perform well, likely because a single gradient descent step lacks the necessary personalization ($c2$). Notably, PAC-PFL outperforms all baselines in terms of prediction accuracy and calibration (see $c1$) due to its strong personalization capability, addressing ($c2$), and inherent regularization, resolving ($c3$). Moreover, it exhibits the best generalization to new clients, overcoming ($c4$). The results for other regression datasets are available in Appendix 8.6.

Test accuracy and CE for existing clients in the classification datasets are reported in Table 3. Remarkably, PAC-PFL outperforms other baselines on the FEMNIST (20) and EMNIST datasets. On FEMNIST (500), pFedGP demonstrates a slight advantage in terms of the mean, albeit with a considerably higher standard deviation, indicating sensitivity to initialization. Given the closely aligned means for pFedGP and PAC-PFL and the fact that the confidence interval of PAC-PFL is encompassed within that of pFedGP, PAC-PFL is a more reliable method. Additionally, pFedGP has a substantially higher computational cost than PAC-PFL[2]. Therefore, we conclude that *PAC-PFL is the superior choice for all datasets.*

---

[2]Achituve et al. (2021) propose alternative variants of pFedGP that trade off accuracy for reduced computational cost. However, we employ the original algorithm.

[4]CE is not calculated for pFedBayes as the available software only provides prediction means and lacks variances required for CE computation.

Table 3: Comparison of probabilistic (▢) and non-probabilistic (▢) FL approaches along with probabilistic non-federated baselines (▢) on classification tasks. Average test accuracy (%) for all baselines (▢, ▢, ▢) and calibration error (CE) for probabilistic approaches (▢, ▢) over 5 trials are reported, where ± captures a 95% confidence interval. [4]Best results (▢) in each column are marked.

| Dataset | FEMNIST (20) | | FEMNIST (500) | | EMNIST | |
|---|---|---|---|---|---|---|
| Metric | Accuracy | CE | Accuracy | CE | Accuracy | CE |
| PAC-PFL | $94.2 \pm 2.6$ | $0.08 \pm 0.01$ | $97.1 \pm 1.6$ | $0.05 \pm 0.01$ | $87.1 \pm 1.1$ | $0.04 \pm 0.01$ |
| pFedGP | $83.6 \pm 2.0$ | $0.10 \pm 0.04$ | $97.7 \pm 6.6$ | $0.02 \pm 0.01$ | $82.4 \pm 1.0$ | $0.06 \pm 0.02$ |
| pFedBayes | $87.0 \pm 2.0$ | - | $88.3 \pm 2.4$ | - | $79.0 \pm 1.3$ | - |
| FedAvg | $88.1 \pm 1.7$ | - | $96.7 \pm 0.5$ | - | $79.9 \pm 1.0$ | - |
| MTL | $75.8 \pm 3.7$ | - | $87.5 \pm 0.1$ | - | $78.2 \pm 1.0$ | - |
| MAML | $82.0 \pm 6.7$ | - | $88.1 \pm 2.6$ | - | $81.5 \pm 2.8$ | - |
| Vanilla | $81.9 \pm 1.3$ | $0.12 \pm 0.01$ | $92.5 \pm 1.8$ | $0.33 \pm 0.06$ | $71.0 \pm 1.2$ | $0.09 \pm 0.01$ |
| Pooled | $89.4 \pm 4.3$ | $0.11 \pm 0.05$ | $94.3 \pm 2.1$ | $0.07 \pm 0.03$ | $63.8 \pm 2.9$ | $0.04 \pm 0.02$ |

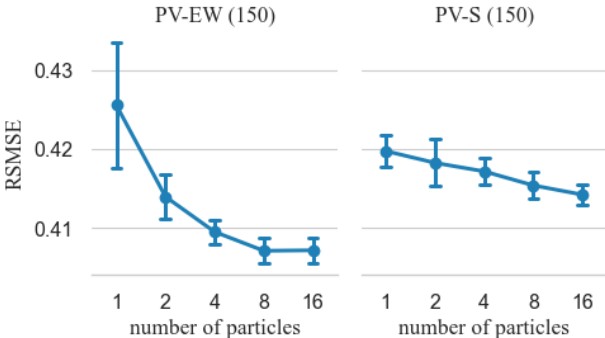

Figure 3: Ablation study on the impact of the number of SVGD particles, $k$, on RSMSE of the existing clients in PV-EW (150) and PV-S (150) datasets. Each experiment is repeated over 5 random seeds. The error bars correspond to the mean ± standard deviation. Computational cost scales linearly with $k$ (see Appendix 8.3.6).

**Ablation study on $k$.** Figure 3 illustrates the impact of the number of SVGD particles, $k$, on RSMSE for existing clients in the PV-EW (150) and PV-S (150) datasets. In both datasets, increasing $k$ enhances the SVGD approximation and the overall performance. The improvement is particularly pronounced in PV-EW due to its higher heterogeneity level and bimodal nature. Notably, transitioning from one to two particles leads to a significant performance boost, as a single particle fails to capture patterns in both modes. This highlights the *efficacy of learning a hyper-posterior instead of a global prior*, corresponding to $k = 1$.

## 7 Conclusion

This paper presents PAC-PFL, a novel PFL algorithm that enables the learning of probabilistic models. The proposed approach learns a shared hyper-posterior in a federated manner, which clients use to sample their priors for personalized posterior inference. To prevent overfitting, PAC-PFL minimizes an upper bound on the true risk of the clients participating in federated training. Moreover, the learned hyper-posterior can be applied to new clients who did not participate in the training, resulting in positive transfer. Conducting experiments on several heterogeneous datasets for regression and classification, we empirically demonstrate that PAC-PFL produces accurate and well-calibrated predictions.

There are two main directions for future research: improving client-level computational complexity (detailed in Appendix 8.3.6) and addressing the privacy-utility trade-off more effectively. Our framework leverages DP to derive valid generalization bounds despite having data-dependent priors and to avoid data leakage, as typical in FL. While our theoretical results in Section 4 show that our ideal pipeline provides DP, we forfeit this property due to the SVGD approximation technique. DP can be reintroduced to prevent data leakage using the common method of injecting noise during training (Geyer et al., 2017), as demonstrated in Appendix 8.4, but this may compromise accuracy (Bagdasaryan et al., 2019). Exploring alternative privacy techniques is an avenue for future research.

**Acknowledgments**

This work was supported as a part of NCCR Automation, a National Centre of Competence in Research, funded by the Swiss National Science Foundation (grant number 51NF40_225155) and the NECON project (grant number 200021219431).

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

# 8 Appendix

## Reproducibility statement

For all theorems and theoretical results, we present detailed assumptions and proofs in Appendix 8.2. Moreover, we provide a comprehensive table containing all parameters utilized throughout the paper in Appendix 8.3.5. This table highlights the interconnections between these parameters and marks the free parameters that can be tuned for optimal utilization of our algorithm.

Regarding the datasets, we employ the FEMNIST dataset, which is curated and maintained by the LEAF project (Caldas et al., 2019). We utilize the original train-test split provided with the data, without any additional preprocessing. The PV dataset can be accessed via the following link: `https://drive.google.com/drive/folders/153MeAlntN4VORHdgYQ3wG3OylWOSlBf9?usp=sharing`.

The source code for our PAC-PFL implementation using GP is accessible within the same Google Drive repository. Upon acceptance, we intend to make the source code for BNN publicly available. To facilitate the use of our software, we have incorporated a demonstration Jupyter Notebook in the source code repository. Furthermore, we have included pre-trained models for PAC-PFL and other baseline models for the PV dataset. Finally, we provide a notebook that generates the figures featured in the paper.

## 8.1 Interpretation through the principle of maximum entropy

In this section, we provide additional justification for the optimal hyper-posterior derived in Corollary 4.4 based on the principle of maximum entropy (Jaynes, 1957). The principle of maximum entropy suggests that when only the class of a distribution is known, the distribution with the highest entropy should be chosen as the least-informative default. The distribution class can be specified by certain moment constraints.This principle is motivated by two key reasons: first, maximizing entropy minimizes the amount of prior information embedded in the distribution, allowing for a more agnostic representation; second, it aligns with the observation that many physical systems tend to evolve towards configurations of maximal entropy over time. While the principle of maximum entropy is commonly employed to derive prior probability distributions in Bayesian inference (Merwe & Skilling, 2010), we utilize it in the context of obtaining the optimal hyper-posterior distribution.

Consider the following constrained maximum entropy problem:

$$\max_{\mathcal{Q}} \ H_{\mathcal{P}}(\mathcal{Q}) \tag{10a}$$

$$s.t. \ -\mathbb{E}_{P \sim \mathcal{Q}} \ln Z_\beta(P, \mathcal{S}_i) + I(\epsilon_i, m_i, \delta) \leq -\mathbb{E}_{P \sim \mathcal{P}} \ln Z_\beta(P, \mathcal{S}_i) \quad \forall i \in \{1, \cdots, n\}. \tag{10b}$$

In Equation (10a), $H_{\mathcal{P}}(\mathcal{Q}) = -KL(\mathcal{Q}\|\mathcal{P})$ represents Jaynes' entropy with the hyper-prior $\mathcal{P}$ serving as the *invariant measure* (Jaynes, 1957). The objective is to maximize this entropy subject to $n$ constraints on the expectations of $\ln Z_\beta(P, \mathcal{S}_i)$ under the distribution $\mathcal{Q}$, as given in (10b). The constants $\epsilon_i$ and $I(\epsilon_i, m_i, \delta)$ are determined by Lemma 4.5 and Theorem 3.2, respectively. Below, we establish a connection between the maximum entropy problem and our approach in Section 4.

**Proposition 8.1.** *The minimizer of the server-level upper bound, $\mathcal{Q}^*$, derived in Corollary 4.4, coincides with the maximizer of the constrained maximum entropy problem presented in (10a)-(10b), when the maximum entropy problem is solved by optimizing the Lagrange function:*

$$\arg\max_{\mathcal{Q}} \ H_{\mathcal{P}}(\mathcal{Q}) + \tau \sum_{i=1}^{n} \Big( \mathbb{E}_{P \sim \mathcal{Q}} \ln Z_\beta(P, \mathcal{S}_i) - I(\epsilon_i, m_i, \delta) - \mathbb{E}_{P \sim \mathcal{P}} \ln Z_\beta(P, \mathcal{S}_i) \Big), \tag{11}$$

*where the Lagrange multiplier $\tau$ is used for all constraints. The constant $\tau$ is as per Corollary 4.4.*

*Proof.* Let $\tilde{\mathcal{Q}}^*$ denote the maximizer of the Lagrange function in (11). By removing the terms in (11) that are constant with respect to $\mathcal{Q}$, we obtain:

$$\tilde{\mathcal{Q}}^* = \arg\max_{\mathcal{Q}} \ H_{\mathcal{P}}(\mathcal{Q}) + \tau \sum_{i=1}^{n} \mathbb{E}_{P \sim \mathcal{Q}} \ln Z_{\beta}(P, \mathcal{S}_i). \tag{12}$$

According to the definition of Jaynes' entropy,

$$\tilde{\mathcal{Q}}^* = \arg\min_{\mathcal{Q}} \ \frac{1}{\tau} KL(\mathcal{Q}\|\mathcal{P}) - \sum_{i=1}^{n} \mathbb{E}_{P \sim \mathcal{Q}} \ln Z_{\beta}(P, \mathcal{S}_i), \tag{13}$$

where we have multiplied the objective in (12) by $-1/\tau$ and changed maximization into minimization. By substituting the formula for $\tau$, one can verify that (13) is equivalent to the server-level upper bound, except for some constant values. Hence, $\tilde{\mathcal{Q}}^* = \mathcal{Q}^*$. $\qquad\square$

The maximum entropy interpretation of $\mathcal{Q}^*$ allows us to analyze the effect of sampling $P$ from $\mathcal{Q}^*$ on client-level bounds. If client $i$ chooses not to participate in FL and decides not to use $\mathcal{Q}^*$, the best alternative approach is to sample $P$ from $\mathcal{P}$. The following corollary provides a comparison between sampling $P$ from $\mathcal{Q}^*$ and sampling $P$ from $\mathcal{P}$.

**Corollary 8.2.** *The $i$-th constraint in (10b) imposes that the expected upper bound for client $i$ is tighter when sampling the prior $P$ from the optimal hyper-posterior, $\mathcal{Q}^*$, compared to sampling from the hyper-prior, $\mathcal{P}$.*

*Proof.* Since a prior which is sampled from the hyper-prior is no longer data-dependent, we utilize the result from Alquier et al. (2016) to derive a bound for client $i$:

$$\mathcal{L}^C(Q_i, \mathcal{D}_i) \leq \hat{\mathcal{L}}^C(Q_i, \mathcal{S}_i) + \frac{1}{\beta}\Big(KL(Q_i\|P) + \frac{\beta^2(b-a)^2}{8m_i} + \ln\big(\frac{1}{\delta}\big)\Big), \tag{14}$$

which holds with probability at least $1-\delta$ over $\mathcal{S}_i \sim \mathcal{D}_i^{m_i}$. The upper bound in (14) is equal to the upper bound in (4) with $\tilde{\mathcal{S}}_i = \emptyset$, except for the constant term $I$. Therefore, the posterior that minimizes the right-hand side of (14) is the same as $\mathbb{Q}^*(P, \mathcal{S}_i)$ derived in Corollary 3.4 with $\tilde{\mathcal{S}}_i = \emptyset$. By plugging $\mathbb{Q}^*(P, \mathcal{S}_i)$ into (14), we obtain the counterpart of (6):

$$\mathcal{L}^C(\mathbb{Q}^*(P, \mathcal{S}_i), \mathcal{D}_i) \leq \frac{1}{\beta}\Big(-\ln Z_{\beta}(P, \mathcal{S}_i) + \frac{\beta^2(b-a)^2}{8m_i} + \ln\big(\frac{1}{\delta}\big)\Big), \tag{15}$$

holding with probability at least $1-\delta$ over $\mathcal{S}_i \sim \mathcal{D}_i^{m_i}$. The $i$-th constraint in (10b) is obtained by taking the expectation of the upper bounds in (6) and (15) when $P \sim \mathcal{Q}^*$ and $P \sim \mathcal{P}$, respectively, and enforcing that the former is smaller than the latter. $\qquad\square$

According to Corollary 8.2, participating in FL is beneficial for client $i$ if the $i$-th constraint is satisfied. However, since we solved the constrained problem per (10a)-(10b) using the Lagrange method with a single multiplier, the constraints might be violated. The constraints are more likely to be satisfied when $\tau$ is large, which can be achieved by having small $n$, $n_2$, and $\beta$, while simultaneously having a large value for $\lambda_E$. In other words, when there are fewer clients participating in FL, the hyper-posterior is more likely to provide improvements for those clients.

*Remark* 8.3. In a similar manner, we can avoid any PAC arguments and use the principle of minimum cross entropy for calculating the posterior. With these two rules, the optimal posterior and hyper-posterior are the same as those obtained by minimizing client-level and server-level PAC bounds.

## 8.2 Proofs and derivations

We first mention without proof a powerful lemma, called the change of measure inequality, that is the basis of proving PAC bounds in most papers. The statement below is adapted from the Appendix of Pentina & Lampert (2014).

**Lemma 8.4** ((Pentina & Lampert, 2014)). *Let $f$ be a random variable taking values in a set $A$ and let $X_1, \cdots, X_l$ be $l$ independent random variables with each $X_k$ distributed according to $\mu_k$ over the set $A_k$. For functions $g_k : A \times A_k \to \mathbb{R}$, $k = 1, \cdots, l$, let $\xi_k(f) = \mathbb{E}_{X_k \sim \mu_k} g_k(f, X_k)$ denote the expectation of $g_k$ under $X_k \sim \mu_k$ as a function of $f$. Then, for any fixed distributions $\pi, \rho$ over $A$ and any $\gamma > 0$, we have that*

$$\mathbb{E}_{f \sim \rho} \Big[ \sum_{k=1}^{l} \big( \xi_k(f) - g_k(f, X_k) \big) \Big] \leq \frac{1}{\gamma} KL(\rho \| \pi) + \frac{1}{\gamma} \psi(\gamma),$$

*where $\psi(\gamma) := \ln \mathbb{E}_{f \sim \pi} \Big[ e^{\gamma \sum_{k=1}^{l} \big( \xi_k(f) - g_k(f, X_k) \big)} \Big]$ is referred to as the log moment-generating function.*

When $\xi_k - g_k$ is bounded, we bound the expectation of the log moment-generating function in Corollary 8.6, which uses the Hoeffding's lemma stated below.

**Lemma 8.5** ((Hoeffding, 1963)). *Let $Y$ be a zero-mean real-valued random variable such that $Y \in [a, b]$ almost surely, i.e. with probability one. Then for any $\gamma > 0$:*

$$\mathbb{E}\big[ e^{\gamma Y} \big] \leq e^{\frac{\gamma^2}{8}(b-a)^2}.$$

**Corollary 8.6.** *If $\xi_k(f) - g_k(f, X_k) \in [a_k, b_k]$ almost surely for all $f$ and $X_k$, it holds for every $\gamma \geq 1$ that $\mathbb{E}_{X_1 \sim \mu_1} \cdots \mathbb{E}_{X_l \sim \mu_l} e^{\frac{1}{\gamma} \psi(\gamma)} \leq e^{\frac{\gamma}{8} \sum_{k=1}^{l} (b_k - a_k)^2}$.*

*Proof of Corollary 8.6.* By taking the expectation of the moment-generating function w.r.t every $X_k$,

$$\mathbb{E}_{X_1 \sim \mu_1} \cdots \mathbb{E}_{X_l \sim \mu_l} e^{\psi(\gamma)} = \mathbb{E}_{X_1 \sim \mu_1} \cdots \mathbb{E}_{X_l \sim \mu_l} \mathbb{E}_{f \sim \pi} \Big[ \prod_{k=1}^{l} e^{\gamma \big( \xi_k(f) - g_k(f, X_k) \big)} \Big]$$

$$= \mathbb{E}_{f \sim \pi} \mathbb{E}_{X_1 \sim \mu_1} \cdots \mathbb{E}_{X_l \sim \mu_l} \Big[ \prod_{k=1}^{l} e^{\gamma \big( \xi_k(f) - g_k(f, X_k) \big)} \Big],$$

where in the last line we have changed the order of expectations. For a given $f$, the terms $\xi_k(f) - g_k(f, X_k)$ for $k \in \{1, \cdots, l\}$ are independent from each other which allows applying Lemma 8.5:

$$\mathbb{E}_{X_1 \sim \mu_1} \cdots \mathbb{E}_{X_l \sim \mu_l} e^{\psi(\gamma)} = \mathbb{E}_{f \sim \pi} \Big[ \prod_{k=1}^{l} \mathbb{E}_{X_k \sim \mu_k} e^{\gamma \big( \xi_k(f) - g_k(f, X_k) \big)} \Big]$$

$$\leq \mathbb{E}_{f \sim \pi} \Big[ \prod_{k=1}^{l} e^{\frac{\gamma^2 (b_k - a_k)^2}{8}} \Big] = e^{\frac{\gamma^2}{8} \sum_{k=1}^{l} (b_k - a_k)^2}.$$

Since $1/\gamma \leq \gamma$, we can use Jensen's inequality (Jensen, 1906) to write:

$$\mathbb{E}_{X_1 \sim \mu_1} \cdots \mathbb{E}_{X_l \sim \mu_l} e^{\frac{1}{\gamma} \psi(\gamma)} \leq e^{\frac{\gamma}{8} \sum_{k=1}^{l} (b_k - a_k)^2}.$$

$\square$

One can utilize Markov's inequality[5] to remove the expectation in Corollary 8.6 and obtain a probabilistic bound on $\psi(\gamma)$. We will use Corollary 8.6 multiple times, and thus, postpone applying Markov's inequality to avoid simultaneous stochastic inequalities which must be combined with a union bound argument.

### 8.2.1 Proof of Theorem 4.2

The proof of Theorem 4.2 is carried out in three steps. The first two steps bound the true risk of clients with $\tilde{\mathcal{S}}_i = \varnothing$ and clients with enlarged datasets, respectively. By merging these two, we will obtain a bound

---

[5]According to Markov's inequality, if $X$ is a nonnegative random variable and $a > 0$, then $\Pr[X \geq a] \leq \mathbb{E}[X]/a$.

on the server-level true risk. In the last step, the closed-form formula of the optimal posterior is exploited to make some simplifications. The main assumptions are independence of clients (guaranteeing independence of $X_k$ in Lemma 8.4), boundedness of the loss function (needed to apply Corollary 8.6), adoption of the optimal posterior (yielding the bound per (6)), and existence of a finite $\epsilon \in \mathbb{R}_+$ such that sampling $P$ from $\mathcal{Q}$ preserves $\epsilon$-DP for all clients (required to use Corollary 3.4).

**Step 1.** We apply Lemma 8.4 with the following instances: take $l = \sum_{i=1}^{n} m_i$ and assign a random variable to each observed sample by clients, $X_k = \mathbf{z}_{ij}$, where $\mathbf{z}_{ij}$ is the $j$'th sample of $\mathcal{S}_i$. Let $\alpha : \{1, \cdots, l\} \to \{1, \cdots, n\}$ be a mapping from each random variable $X_k$ to the corresponding client, $\alpha(k) = i$ if $X_k = \mathbf{z}_{ij}$ for some $j$. Correspondingly, we take $\mu_k = \mathcal{D}_{\alpha(k)}$ to be the respective distribution. Further, we set $f = (P, h_1, \cdots, h_n)$ to be a tuple of one prior and $n$ hypotheses and consider distributions $\pi = (\mathcal{P}, P, \cdots, P)$ and $\rho = (\mathcal{Q}, \mathbb{Q}^*(P, \mathcal{S}_1), \cdots, \mathbb{Q}^*(P, \mathcal{S}_n))$ over it. Each function $g_k$ is designated to be one of the summands in the empirical server-level risk, $g_k(f, X_k) = \frac{1}{nm_{\alpha(k)}} \ell(h_{\alpha(k)}, X_k)$. By invoking Lemma 8.4 with $\gamma = \lambda_1 \geq 1$, we have:

$$\frac{1}{n} \mathbb{E}_{P \sim \mathcal{Q}} \sum_{i=1}^{n} \mathcal{L}^C(\mathbb{Q}^*(P, \mathcal{S}_i), \mathcal{D}_i) \leq \frac{1}{n} \mathbb{E}_{P \sim \mathcal{Q}} \sum_{i=1}^{n} \hat{\mathcal{L}}^C(\mathbb{Q}^*(P, \mathcal{S}_i), \mathcal{S}_i) \tag{*}$$

$$+ \frac{1}{\lambda_1} KL(\mathcal{Q}\|\mathcal{P}) + \frac{1}{\lambda_1} \sum_{i=1}^{n} \mathbb{E}_{P \sim \mathcal{Q}} KL(\mathbb{Q}^*(P, \mathcal{S}_i)\|P) \tag{$\dagger$}$$

$$+ \frac{1}{\lambda_1} \psi_1(\lambda_1). \tag{16}$$

Line ($\dagger$) is equal to $KL(\rho\|\pi)$ due to (13-15) in Rothfuss et al. (2021), and $\psi_1(\lambda_1)$ is a log moment-generating function defined as:

$$\psi_1(\lambda_1) := \ln \mathbb{E}_{P \sim \mathcal{P}} \mathbb{E}_{h \sim P} e^{\frac{\lambda_1}{n} \sum_{i=1}^{n} \left( \mathbb{E}_{\mathbf{z} \sim \mathcal{D}_i} \ell(h, \mathbf{z}) - \frac{1}{m_i} \sum_{\mathbf{z} \in \mathcal{S}_i} \ell(h, \mathbf{z}) \right)}.$$

For $\ell \in [a, b]$, we apply Corollary 8.6 acknowledging that $|\xi_k - g_k| \leq (b-a)/nm_{\alpha(k)}$, hence, obtaining:

$$\mathbb{E}_{\mathcal{S}_1 \sim \mathcal{D}_1^{m_1}} \cdots \mathbb{E}_{\mathcal{S}_n \sim \mathcal{D}_n^{m_n}} e^{\frac{1}{\lambda_1} \psi_1(\lambda_1)} \leq e^{\frac{\lambda_1}{8n^2} (b-a)^2 \sum_{i=1}^{n} \frac{1}{m_i}}. \tag{17}$$

The right-hand side of (*) matches the server-level empirical loss per (8), but the left-hand side is different from the true risk in (7), as new clients and new samples are missing.

**Step 2.** In the second step, we use $l = n$ and assign one random variable $X_k$ to each client. To maintain a cohesive notation, we will use subscript $i$ instead of $k$ for elements of Lemma 8.4. We substitute each $X_i$ with a subset of $\tilde{m}_i$ samples from $\mathcal{S}_i$ drawn without replacement, which is possible as $\tilde{m}_i \leq m_i$. Set $f = P$, $\pi = \mathcal{P}$, $\rho = \mathcal{Q}$, and $g_i(f, X_i) = \frac{1}{n} \mathcal{L}^C(\mathbb{Q}^*(f, \mathcal{S}_i \cup X_i), \mathcal{D}_i)$. Notice that $\mathcal{S}_i$ and $\mathcal{D}_i$ are embedded in the definition of $g_i$ and not given as function arguments. Let $\lambda_2 = \lambda/(n_2 + \upsilon) \geq 1$, where $n_2$ is the number of clients with $\tilde{m}_i > 0$ and $\upsilon$ is a small positive number. By applying Lemma (8.4) with parameter $\gamma = \lambda_2$, we obtain:

$$\frac{1}{n} \mathbb{E}_{P \sim \mathcal{Q}} \sum_{i=1}^{n} \mathbb{E}_{\tilde{\mathcal{S}}_i \sim \mathcal{D}_i^{\tilde{m}_i}} \mathcal{L}^C(\mathbb{Q}^*(P, \mathcal{S}_i \cup \tilde{\mathcal{S}}_i), \mathcal{D}_i) \leq \frac{1}{n} \mathbb{E}_{P \sim \mathcal{Q}} \sum_{i=1}^{n} \mathcal{L}^C(\mathbb{Q}^*(P, \mathcal{S}_i), \mathcal{D}_i) \tag{$\diamond$}$$

$$+ \frac{1}{\lambda_2} KL(\mathcal{Q}\|\mathcal{P}) + \frac{1}{\lambda_2} \psi_2(\lambda_2), \tag{18}$$

$$\psi_2(\lambda_2) := \ln \mathbb{E}_{P \sim \mathcal{P}} e^{\frac{\lambda_2}{n} \sum_{i=1}^{n} \left( \mathbb{E}_{\tilde{\mathcal{S}}_i \sim \mathcal{D}_i^{\tilde{m}_i}} \mathcal{L}^C\left(\mathbb{Q}^*(P, \mathcal{S}_i \cup \tilde{\mathcal{S}}_i), \mathcal{D}_i\right) - \mathcal{L}^C\left(\mathbb{Q}^*(P, \mathcal{S}_i), \mathcal{D}_i\right) \right)}.$$

When $\tilde{m}_i = 0$ for all clients, the two sides of ($\diamond$) are equal, $\psi_2(\lambda_2)$ is zero, and the weight of the KL term in (18) goes to zero as $\upsilon \to 0$. Thus, by setting $\lambda_2$ proportional to $n_2 + \upsilon$, (16) and (18) are consistent.

It is evident from the definition of $\xi_i$ and $g_i$ that $|\xi_i - g_i| \le (b-a)/n$. Still, we can exploit the explicit formula of $\mathbb{Q}^*$ per (5) to obtain a potentially smaller range by controlling the effect of a new sample of size $\tilde{m}_i$ on the optimal posterior. Let $Q_i^* = \mathbb{Q}^*(P, \mathcal{S}_i)$ and $\tilde{Q}_i^* = \mathbb{Q}^*(P, \mathcal{S}_i \cup \tilde{\mathcal{S}}_i)$ for some fixed $P$ and $\tilde{\mathcal{S}}_i$. From the closed form formula of $\mathbb{Q}^*$ per (5), for every $h \in \mathcal{H}$:

$$\frac{\tilde{Q}_i^*(h)}{Q_i^*(h)} = \frac{e^{\frac{-\beta}{m_i + \tilde{m}_i} \sum_{\mathbf{z} \in \mathcal{S}_i \cup \tilde{\mathcal{S}}_i} \ell(h, \mathbf{z})}}{e^{\frac{-\beta}{m_i} \sum_{\mathbf{z} \in \mathcal{S}_i} \ell(h, \mathbf{z})}} \cdot \frac{\mathbb{E}_{h \sim P} e^{\frac{-\beta}{m_i} \sum_{\mathbf{z} \in \mathcal{S}_i} \ell(h, \mathbf{z})}}{\mathbb{E}_{h \sim P} e^{\frac{-\beta}{m_i + \tilde{m}_i} \sum_{\mathbf{z} \in \mathcal{S}_i \cup \tilde{\mathcal{S}}_i} \ell(h, \mathbf{z})}} \tag{19}$$

From the bounded loss assumption, we have:

$$\Big| \frac{1}{m_i + \tilde{m}_i} \sum_{\mathbf{z} \in \mathcal{S}_i \cup \tilde{\mathcal{S}}_i} \ell(h, \mathbf{z}) - \frac{1}{m_i} \sum_{\mathbf{z} \in \mathcal{S}_i} \ell(h, \mathbf{z}) \Big| \le \frac{\tilde{m}_i(b-a)}{m_i + \tilde{m}_i}. \tag{20}$$

From (19) and (20), $\frac{\tilde{Q}_i^*(h)}{Q_i^*(h)} \in \big[ e^{\frac{-2\beta \tilde{m}_i}{m_i + \tilde{m}_i}(b-a)}, e^{\frac{2\beta \tilde{m}_i}{m_i + \tilde{m}_i}(b-a)} \big]$, which obtains a bound on $\xi_i(f) - g_i(f, X_i)$:

$$\begin{aligned}
\xi_i(f) - g_i(f, X_i) &= \frac{1}{n} \mathbb{E}_{\tilde{\mathcal{S}}_i \sim \mathcal{D}_i^{\tilde{m}_i}} \mathcal{L}^C\big( \mathbb{Q}^*(P, \mathcal{S}_i \cup \tilde{\mathcal{S}}_i), \mathcal{D}_i \big) - \frac{1}{n} \mathcal{L}^C\big( \mathbb{Q}^*(P, \mathcal{S}_i), \mathcal{D}_i \big) \\
&= \frac{1}{n} \mathbb{E}_{\tilde{\mathcal{S}}_i \sim \mathcal{D}_i^{\tilde{m}_i}} \mathbb{E}_{\mathbf{z} \sim \mathcal{D}_i} \int_{\mathcal{H}} \ell(h, \mathbf{z})(\tilde{Q}_i^*(h) - Q_i^*(h)) d_h \\
&\in \frac{1}{n} \mathcal{L}^C\big( \mathbb{Q}^*(P, \mathcal{S}_i), \mathcal{D}_i \big) \cdot \big[ e^{\frac{-2\beta \tilde{m}_i}{m_i + \tilde{m}_i}(b-a)} - 1, e^{\frac{2\beta \tilde{m}_i}{m_i + \tilde{m}_i}(b-a)} - 1 \big] \\
&\subset \frac{b}{n} \big[ \big( e^{\frac{-2\beta \tilde{m}_i}{m_i + \tilde{m}_i}(b-a)} - 1 \big), \big( e^{\frac{2\beta \tilde{m}_i}{m_i + \tilde{m}_i}(b-a)} - 1 \big) \big]. 
\end{aligned} \tag{21}$$

Comparing (21) and the naive inequality $|\xi_i - g_i| \le (b-a)/n$, it can be verified that $|\xi_i - g_i| \le \Delta_i$, where:

$$\Delta_i := \frac{1}{n} \min \big\{ b-a, b\big( e^{\frac{2\beta \tilde{m}_i}{m_i + \tilde{m}_i}(b-a)} - e^{\frac{-2\beta \tilde{m}_i}{m_i + \tilde{m}_i}(b-a)} \big) \big\}.$$

It is possible to obtain a tighter range, $\Delta_i < (b-a)/n$, when $m_i$ is large, $\tilde{m}_i$ is small, or $\beta$ is small. Particularly, for $\tilde{m}_i = 0$, $\Delta_i = 0$ but $b-a$ provides a vacuous bound.

By applying Corollary 8.6, one obtains:

$$\mathbb{E}_{\tilde{\mathcal{S}}_1 \sim \mathcal{D}_1^{\tilde{m}_1}} \cdots \mathbb{E}_{\tilde{\mathcal{S}}_n \sim \mathcal{D}_n^{\tilde{m}_n}} e^{1/\lambda_2 \psi_2(\lambda_2)} \le e^{\frac{\lambda_2}{8} \sum_{i=1}^n \Delta_i^2} \le e^{\frac{\lambda_2}{8n}(b-a)^2}. \tag{22}$$

Note that $\mathcal{S}_i \ne \varnothing$ is required for defining $X_i$. Hence, new clients are not added to the analysis yet.

**Merging steps 1 and 2.** Bringing (16) and (18) together, we get:

$$\begin{aligned}
\mathcal{L}^S(\mathcal{Q}, \mathcal{D}, \mathcal{S}, \tilde{\mathbf{m}}) \le &\hat{\mathcal{L}}^S(\mathcal{Q}, \mathcal{S}) + \big( \frac{1}{\lambda_1} + \frac{1}{\lambda_2} \big) KL(\mathcal{Q} \| \mathcal{P}) + \frac{1}{\lambda_1} \sum_{i=1}^n \mathbb{E}_{P \sim \mathcal{Q}} KL\big( \mathbb{Q}^*(P, \mathcal{S}_i) \| P \big) \\
&+ \frac{1}{\lambda_1} \psi_1(\lambda_1) + \frac{1}{\lambda_2} \psi_2(\lambda_2).
\end{aligned} \tag{23}$$

Equation (23) provides a bound on the server-level true loss and incorporates various components such as the empirical loss, complexity penalty terms (represented by KL divergences between the posterior and the prior, and between the hyper-posterior and the hyper-prior), and two log moment-generating functions.

Next, we bound the weighted sum of the log moment-generating functions in (23) when $\ell \in [a, b]$. One obtains from Markov's inequality that:

$$Pr \Big[ e^{\frac{1}{\lambda_1} \psi_1(\lambda_1) + \frac{1}{\lambda_2} \psi_2(\lambda_2)} \le \frac{1}{\delta} e^{\frac{\lambda_1 (b-a)^2}{8n^2} \sum_{i=1}^n \frac{1}{m_i} + \frac{\lambda_2}{8} \sum_{i=1}^n \Delta_i^2} \Big] \ge$$

$$1 - \frac{\mathbb{E}_{\mathcal{S}_1 \sim \mathcal{D}_1^{m_1}} \cdots \mathbb{E}_{\mathcal{S}_n \sim \mathcal{D}_n^{m_n}} \mathbb{E}_{\tilde{\mathcal{S}}_1 \sim \mathcal{D}_1^{\tilde{m}_1}} \cdots \mathbb{E}_{\tilde{\mathcal{S}}_n \sim \mathcal{D}_n^{\tilde{m}_n}} e^{\frac{1}{\lambda_1} \psi_1(\lambda_1) + \frac{1}{\lambda_2} \psi_2(\lambda_2)}}{\frac{1}{\delta} e^{\frac{\lambda_1 (b-a)^2}{8n^2} \sum_{i=1}^n \frac{1}{m_i} + \frac{\lambda_2}{8} \sum_{i=1}^n \Delta_i^2}}, \tag{24}$$

where the probability is taken over $\mathcal{S}_i \sim \mathcal{D}_i^{m_i}$ and $\tilde{\mathcal{S}}_i \sim \mathcal{D}_i^{\tilde{m}_i}$ for $i = 1, \cdots, n$ and $\delta \in (0, 1)$ is the confidence level given in Theorem 4.2. To bound the right-hand side, we multiply (17) and (22), which is possible as they involve expectations over independent random variables, resulting in:

$$\mathbb{E}_{\mathcal{S}_1 \sim \mathcal{D}_1^{m_1}} \cdots \mathbb{E}_{\mathcal{S}_n \sim \mathcal{D}_n^{m_n}} \mathbb{E}_{\tilde{\mathcal{S}}_1 \sim \mathcal{D}_1^{\tilde{m}_1}} \cdots \mathbb{E}_{\tilde{\mathcal{S}}_n \sim \mathcal{D}_n^{\tilde{m}_n}} e^{\frac{1}{\lambda_1}\psi_1(\lambda_1) + \frac{1}{\lambda_2}\psi_2(\lambda_2)} \leq e^{\frac{\lambda_1(b-a)^2}{8n^2} \sum_{i=1}^n \frac{1}{m_i} + \frac{\lambda_2}{8} \sum_{i=1}^n \Delta_i^2}. \tag{25}$$

By putting together (24) and (25), we derive:

$$Pr\Big[\frac{1}{\lambda_1}\psi_1(\lambda_1) + \frac{1}{\lambda_2}\psi_2(\lambda_2) \leq \frac{\lambda_1(b-a)^2}{8n^2} \sum_{i=1}^n \frac{1}{m_i} + \frac{\lambda_2}{8} \sum_{i=1}^n \Delta_i^2 + \ln(\frac{1}{\delta})\Big] =$$
$$Pr\Big[e^{\frac{1}{\lambda_1}\psi_1(\lambda_1) + \frac{1}{\lambda_2}\psi_2(\lambda_2)} \leq \frac{1}{\delta} e^{\frac{\lambda_1(b-a)^2}{8n^2} \sum_{i=1}^n \frac{1}{m_i} + \frac{\lambda_2}{8} \sum_{i=1}^n \Delta_i^2}\Big] \geq 1 - \delta, \tag{26}$$

where the probability is taken over $\mathcal{S}_i \sim \mathcal{D}_i^{m_i}$ and $\tilde{\mathcal{S}}_i \sim \mathcal{D}_i^{\tilde{m}_i}$ for $i = 1, \cdots, n$.

One obtains from (23) and (26) that:

$$Pr\Big[\mathcal{L}^S(\mathcal{Q}, \mathcal{D}, \mathcal{S}, \tilde{\mathbf{m}}) \leq \hat{\mathcal{L}}^S(\mathcal{Q}, \mathcal{S}) + \Big(\frac{1}{\lambda_1} + \frac{1}{\lambda_2}\Big) KL(\mathcal{Q}\|\mathcal{P}) + \frac{1}{\lambda_1} \sum_{i=1}^n \mathbb{E}_{P \sim \mathcal{Q}} KL\big(\mathbb{Q}^*(P, \mathcal{S}_i)\|P\big)$$
$$+ \frac{\lambda_1(b-a)^2}{8n^2} \sum_{i=1}^n \frac{1}{m_i} + \frac{\lambda_2}{8} \sum_{i=1}^n \Delta_i^2 + \ln\big(\frac{1}{\delta}\big)\Big] \geq 1 - \delta. \tag{27}$$

**Step 3.** In the final step, we utilize the closed-form expression of the optimal posterior for each client, $\mathbb{Q}^*$, based on Corollary 3.4, to simplify (27). By substituting the definition of the server-level empirical loss given in (8) into (27) and refactoring some terms, one obtains:

$$Pr\Big[\mathcal{L}^S(\mathcal{Q}, \mathcal{D}, \mathcal{S}, \tilde{\mathbf{m}}) \leq \mathbb{E}_{P \sim \mathcal{Q}} \frac{1}{n} \sum_{i=1}^n \big(\hat{\mathcal{L}}^C(\mathbb{Q}^*(P, \mathcal{S}_i), \mathcal{S}_i) + \frac{n}{\lambda_1} KL\big(\mathbb{Q}^*(P, \mathcal{S}_i)\|P\big)\big) \tag{‡}$$
$$+ \Big(\frac{1}{\lambda_1} + \frac{1}{\lambda_2}\Big) KL(\mathcal{Q}\|\mathcal{P})$$
$$+ \frac{\lambda_1(b-a)^2}{8n^2} \sum_{i=1}^n \frac{1}{m_i} + \frac{\lambda_2}{8} \sum_{i=1}^n \Delta_i^2 + \ln\big(\frac{1}{\delta}\big)\Big] \geq 1 - \delta.$$

When we set $\lambda_1 = n\beta$, each term in the summation in (‡) becomes equivalent to the upper bound for client $i$ in Theorem 3.2 up to a constant. This choice of $\lambda_1$ enables us to simplify (‡) similar to (6), resulting in:

$$Pr\Big[\mathcal{L}^S(\mathcal{Q}, \mathcal{D}, \mathcal{S}, \tilde{\mathbf{m}}) \leq \frac{-1}{n\beta} \sum_{i=1}^n \mathbb{E}_{P \sim \mathcal{Q}} \ln Z_\beta^C(P, \mathcal{S}_i) + \Big(\frac{1}{n\beta} + \frac{1}{\lambda_2}\Big) KL(\mathcal{Q}\|\mathcal{P})$$
$$+ \frac{\beta(b-a)^2}{8n} \sum_{i=1}^n \frac{1}{m_i} + \frac{\lambda_2}{8} \sum_{i=1}^n \Delta_i^2 + \ln\big(\frac{1}{\delta}\big)\Big] \geq 1 - \delta.$$

Note that the selection of $\lambda_1 = n\beta$ is only feasible when $\beta \geq 1/n$ because the first step of this proof requires $\lambda_1 \geq 1$. This is the reason behind the assumption $\beta \geq 1/n$ in Theorem 4.2. By substituting $\lambda_2 = \lambda/(n_2 + v)$, which was motivated in the second step, we get the desired bound. Finally, we use the factorization proposed in Rothfuss et al. (2021) to convert $\ln(1/\delta)$ into $\ln(1/\delta)/\sqrt{n}$.

### 8.2.2 Proof of Lemma 4.3

Assume $\tilde{m}_i$ is only known when $i \in A$, where $A \subset \{1, \cdots, n\}$ is a subset of clients, but is unknown for the rest of the clients, $i \in B = \{1, \cdots, n\} \backslash A$. In an extreme case, $A = \varnothing$ means that $\tilde{m}_i$ is unknown for all clients. Let $\rho(A)$ and $\rho(B)$ denote the number of clients with $\tilde{m}_i > 0$ in sets $A$ and $B$ respectively. It is

clear from the definition that $\rho(A) + \rho(B) = n_2$, where $\rho(A)$ is known but $\rho(B)$ and $n_2$ are unknown. Also, $\rho(A) \leq |A|$ and $\rho(B) \leq |B|$, where $|A|$ and $|B|$ stand for the cardinality of the respective set.

We prove that the upper bound in Theorem 4.2 is looser when replacing $\Delta_i$ with $(b-a)/n$ for $i \in B$ and $n_2$ with $\rho(A) + |B|$. It is enough to show:

$$\frac{\rho(A) + \rho(B) + \upsilon}{\lambda} KL(\mathcal{Q}\|\mathcal{P}) + \frac{\sum_{i \in A} \Delta_i^2 + \sum_{i \in B} \Delta_i^2}{8(\rho(A) + \rho(B) + \upsilon)}\lambda \leq$$
$$\frac{\rho(A) + |B| + \upsilon}{\lambda} KL(\mathcal{Q}\|\mathcal{P}) + \frac{\sum_{i \in A} \Delta_i^2 + |B|(\frac{b-a}{n})^2}{8(\rho(A) + |B| + \upsilon)}\lambda.$$

Since $\rho(B) \leq |B|$ and KL divergence is non-negative, we will prove that:

$$\frac{\sum_{i \in A} \Delta_i^2 + \sum_{i \in B} \Delta_i^2}{\rho(A) + \rho(B) + \upsilon} \leq \frac{\sum_{i \in A} \Delta_i^2 + |B|(\frac{b-a}{n})^2}{\rho(A) + |B| + \upsilon}. \tag{28}$$

For clients in $B$ with $\tilde{m}_i = 0$, $\Delta_i = 0$, and for the rest, $\Delta_i \leq (b-a)/n$. Hence, $\sum_{i \in B} \Delta_i^2 \leq \rho(B)\big((b-a)/n\big)^2$ and a stronger condition than (28) is:

$$\frac{\sum_{i \in A} \Delta_i^2 + \rho(B)(\frac{b-a}{n})^2}{\rho(A) + \rho(B) + \upsilon} \leq \frac{\sum_{i \in A} \Delta_i^2 + |B|(\frac{b-a}{n})^2}{\rho(A) + |B| + \upsilon} \iff \sum_{i \in A} \Delta_i^2 \leq \big(\rho(A) + \upsilon\big)\big(\frac{b-a}{n}\big)^2,$$

which holds since $\sum_{i \in A} \Delta_i^2 \leq \rho(A)\big((b-a)/n\big)^2$ and $\upsilon \geq 0$.

### 8.2.3 Proof of Corollary 4.4

The optimal hyper-posterior minimizes the upper bound on the true server-level risk established in 9. By utilizing the structural similarity between the server-level and client-level upper-bounds, equations (9) and (4), we employ Corollary 3.4 to obtain $\mathcal{Q}^*$.

### 8.2.4 Proof of Lemma 4.6

We invoke Lemma 8.4 by defining $X_i = (\mathcal{D}_i, m_i, \mathcal{S}_i)$ and using $X_i = (X_i[\mathcal{D}], X_i[m], X_i[\mathcal{S}])$ to distinguish between components in $X_i$. The distribution over $X_i$ is $\mu_i = (\mathcal{T}, X_i[\mathcal{D}]^{X_i[m]})$. Additionally, $l = n$, $f = P$, $\pi = \mathcal{P}$, $\rho = \mathcal{Q}^*$, and $g_i(f, X_i) = \frac{1}{n}\mathcal{L}^C\big(\mathbb{Q}^*(f, X_i[\mathcal{S}]), X_i[\mathcal{D}]\big)$ are used. While $g_i$ defined above resembles the one from the second step of Proposition 8.2.1, the difference lies in considering $\mathcal{S}_i$ and $\mathcal{D}_i$ as part of the random variable $X_i$, giving rise to their presence in the expectation. By applying Lemma (8.4) with $\gamma = \tilde{\lambda} \geq 1$, one obtains:

$$\mathbb{E}_{(\mathcal{D}_\iota, \tilde{m}_\iota) \sim \mathcal{T}} \mathbb{E}_{\tilde{\mathcal{S}}_\iota \sim \mathcal{D}_\iota^{\tilde{m}_\iota}} \mathbb{E}_{P \sim \mathcal{Q}^*} \mathcal{L}^C\big(\mathbb{Q}^*(P, \tilde{\mathcal{S}}_\iota), \mathcal{D}_\iota\big) \leq \frac{1}{n}\mathbb{E}_{P \sim \mathcal{Q}^*} \sum_{i=1}^{n} \mathcal{L}^C\big(\mathbb{Q}^*(P, \mathcal{S}_i), \mathcal{D}_i\big)$$
$$+ \frac{1}{\tilde{\lambda}} KL\big(\mathcal{Q}^*\|\mathcal{P}\big) + \frac{1}{\tilde{\lambda}}\tilde{\psi}(\tilde{\lambda}). \tag{29}$$

The left-hand side of (29) is the true loss for a generic new client, $\iota$, sampled from $\mathcal{T}$. The moment generating function and the bound on its expectation due to Corollary 8.6 are:

$$\tilde{\psi}(\tilde{\lambda}) := \ln \mathbb{E}_{P \sim \mathcal{P}} e^{\tilde{\lambda}\left(\mathbb{E}_{(\mathcal{D}_\iota, \tilde{m}_\iota) \sim \mathcal{T}} \mathbb{E}_{\tilde{\mathcal{S}}_\iota \sim \mathcal{D}_\iota^{\tilde{m}_\iota}} \mathcal{L}^C\left(\mathbb{Q}^*(P, \tilde{\mathcal{S}}_\iota), \mathcal{D}_\iota\right) - \frac{1}{n}\sum_{i=1}^{n} \mathcal{L}^C\left(\mathbb{Q}^*(P, \mathcal{S}_i), \mathcal{D}_i\right)\right)},$$

$$\mathbb{E}_{(\mathcal{D}_1, m_1) \sim \mathcal{T}} \cdots \mathbb{E}_{(\mathcal{D}_n, m_n) \sim \mathcal{T}} \mathbb{E}_{\mathcal{S}_1 \sim \mathcal{D}_1^{m_1}} \cdots \mathbb{E}_{\mathcal{S}_n \sim \mathcal{D}_n^{m_n}} e^{\frac{1}{\tilde{\lambda}}\tilde{\psi}(\tilde{\lambda})} \leq e^{\frac{\tilde{\lambda}}{8n}(b-a)^2}. \tag{30}$$

Merging (6) and the result of Step 1 in the proof of Proposition 8.2.1 when $\lambda_1 = 1/(n\beta)$ and $\mathbb{Q}^*$ is the optimal posterior mapping in. 5, we rewrite (29) as:

$$\mathbb{E}_{\mathcal{D}_\iota \sim \mathcal{T}} \mathbb{E}_{\tilde{\mathcal{S}}_\iota \sim \mathcal{D}_\iota^{\tilde{m}_\iota}} \mathbb{E}_{P \sim \mathcal{Q}^*} \mathcal{L}^C \big( \mathbb{Q}^*(P, \tilde{\mathcal{S}}_\iota), \mathcal{D}_\iota \big) \leq \frac{-1}{n\beta} \sum_{i=1}^n \mathbb{E}_{P \sim \mathcal{Q}^*} \ln Z_\beta(P, \mathcal{S}_i)$$
$$+ \big( \frac{1}{\tilde{\lambda}} + \frac{1}{n\beta} \big) KL \big( \mathcal{Q}^* \| \mathcal{P} \big) + \frac{1}{\tilde{\lambda}} \tilde{\psi}(\tilde{\lambda}) + \frac{1}{n\beta} \psi_1(n\beta).$$

Plugging in the formula of $\mathcal{Q}^*$ given in Corollary (4.4) and setting $\tilde{\lambda} = \lambda/(n_2 + \upsilon)$,

$$\mathbb{E}_{\mathcal{D}_\iota \sim \mathcal{T}} \mathbb{E}_{\tilde{\mathcal{S}}_\iota \sim \mathcal{D}_\iota^{\tilde{m}_\iota}} \mathbb{E}_{P \sim \mathcal{Q}^*} \mathcal{L}^C \big( \mathbb{Q}^*(P, \tilde{\mathcal{S}}_\iota), \mathcal{D}_\iota \big) \leq -\big( \frac{1}{n\beta} + \frac{n_2 + \upsilon}{\lambda} \big) \ln Z_\tau^S(\mathcal{P}, \mathcal{S})$$
$$+ \frac{1}{n\beta} \psi_1(n\beta) + \frac{n_2 + \upsilon}{\lambda} \tilde{\psi} \big( \frac{\lambda}{n_2 + \upsilon} \big).$$

Using a similar technique to the proof of Proposition 8.2.1, along with (17) and (30), we obtain the desired result.

### 8.2.5 Looser upper bound for new clients

We consider Theorem 4.2 when $\mathcal{Q} = \mathcal{Q}^*$ and simplify the terms in the upper bound that involve $\mathcal{Q}^*$:

$$\frac{-1}{n\beta} \sum_{i=1}^n \mathbb{E}_{P \sim \mathcal{Q}^*} \ln Z_\beta^C(P, \mathcal{S}_i) + \big( \frac{1}{n\beta} + \frac{n_2 + \upsilon}{\lambda} \big) KL \big( \mathcal{Q}^* \| \mathcal{P} \big)$$
$$= \frac{1}{n\beta\tau} \mathbb{E}_{P \sim \mathcal{Q}^*} \big( -\tau \sum_{i=1}^n \ln Z_\beta^C(P, \mathcal{S}_i) + \ln \frac{\mathcal{Q}^*(P)}{\mathcal{P}(P)} \big) \qquad (31)$$
$$= -\frac{1}{n\beta\tau} \ln Z_\tau^S(\mathcal{P}, \mathcal{S}) = -\big( \frac{1}{n\beta} + \frac{n_2 + \upsilon}{\lambda} \big) \ln Z_\tau^S(\mathcal{P}, \mathcal{S}). \qquad (32)$$

Equation (31) follows from the KL divergence definition and (32) is obtained by substituting the closed-form formula of $\mathcal{Q}^*$ derived in Corollary 4.4 into the expression. To compare the bound in Theorem 4.2 with Lemma 4.6, we can substitute the first two terms in Theorem 4.2 with (32). In order for the bound in Lemma 4.6 to be looser than this substituted bound, it is sufficient to show that $\sum_{i=1}^n \Delta_i^2 \leq (b-a)^2/n$, which is true based on the definition of $\Delta_i$ in Theorem 4.2.

## 8.3 Details of the algorithm

### 8.3.1 Asymptotic behavior and non-vacuousness of bounds

We analyze the asymptotic behavior of the client-level and server-level bounds as the number of samples per client and the number of existing clients approach infinity, i.e., $m_i \to \infty$ and $n \to \infty$. A PAC bound is considered consistent when the gap between the true and empirical risks goes to zero. The client-level and server-level bounds are consistent if and only if: a) $\beta \in \Omega(1)$, b) $\beta \in o(m_i)$, c) $\lambda \in \Omega(n_2)$, d) $\lambda \in o(n(n_2 + \upsilon))$. Here, $o$ and $\Omega$ represent the small-oh and big-omega notations for function growth rate (Cormen et al., 2022).

Non-vacuous bounds are essential, ensuring that the terms independent of the posterior or the hyper-posterior are not excessively large such that the bound on the true loss holds regardless of the empirical loss. To achieve a non-vacuous bound, it is necessary (but not sufficient) that $b - a < 8$ and that $\epsilon_i < \sqrt{2(b-a)}$. The latter condition can be converted into an upper bound on $\lambda$ based on the results from Lemma 4.5.

In our experiments, we follow Rothfuss et al. (2021) and set $\beta = m_i + \tilde{m}_i$, leading to a non-vanishing gap between the true and empirical risks at the client and server levels. However, this choice simplifies the computations as discussed in Section 5. Moreover, this choice leads to faster decay of the KL term in both the client-level and server-level bounds, which can be advantageous when $m_i$ is small (Rothfuss et al., 2021). We tune $\lambda$, while respecting the non-vacuousness condition, to manipulate the regularization strength of the hyper-posterior towards the hyper-prior.

### 8.3.2 Background on GP

Below, we provide additional details regarding the GP models we use. For a more comprehensive overview of GPs, please refer to Rasmussen & Williams (2005). In the rest of this section, we express the dataset of client $i$ as $\mathcal{S}_i = (\mathbf{X}_i, \mathbf{y}_i)$, where $\mathbf{X}_i \in R_{\mathbf{x}}^{m_i} \subset \mathbb{R}^{m_i \times d}$ is a matrix with each data sample as a row. The corresponding target values are stored in the vector $\mathbf{y}_i \in R_y^{m_i} \subset \mathbb{R}^{m_i}$.

**GP with a deep mean and a deep kernel (Wilson et al., 2016).** Let $P_{\boldsymbol{\phi}}(h) = \mathcal{GP}(h|m_{\boldsymbol{\phi}}, k_{\boldsymbol{\phi}})$ denote a GP prior specified by a deep mean function, $m_{\boldsymbol{\phi}}$, a deep kernel function, $k_{\boldsymbol{\phi}}$, and a Gaussian likelihood with noise standard deviation, $\sigma_{\boldsymbol{\phi}} \in \mathbb{R}_+$. The vector $\boldsymbol{\phi} \in \mathbb{R}^{d_{\boldsymbol{\phi}}}$ concatenates all learnable hyper-parameters of the GP prior, including the mean parameters, kernel parameters, and the likelihood noise. The mean function, $m_{\boldsymbol{\phi}} : R_{\mathbf{x}} \to \mathbb{R}$, is implemented as a multi-layer NN with weights given by $\boldsymbol{\phi}$, hyperbolic tangent activation functions in the hidden layers, and linear output functions in the output layer. The kernel function is a squared-exponential (SE) kernel applied on top of an NN, defined as:

$$k_{\boldsymbol{\phi}}(\mathbf{x}, \mathbf{x}') := exp(\frac{-1}{2}\|f_{\boldsymbol{\phi}}(\mathbf{x}) - f_{\boldsymbol{\phi}}(\mathbf{x}')\|_2^2) \in [0, 1] \qquad \forall \mathbf{x} \in R_{\mathbf{x}}, \forall \mathbf{x}' \in R_{\mathbf{x}}.$$

The function $f_{\boldsymbol{\phi}} : R_{\mathbf{x}} \to \mathbb{R}^{d_f}$ represents an NN with weights given by $\boldsymbol{\phi}$ that maps the typically high-dimensional feature vector, $\mathbf{x} \in R_{\mathbf{x}} \subset \mathbb{R}^d$, to a lower-dimensional output vector in $\mathbb{R}^{d_f}$, where in our specific case, $d_f$ is two. Deep kernels serve as feature extractors and allow for learning more sophisticated representations from the data (Ober et al., 2021). The length scale of the SE kernel is set to 1 because the weights of the output layer of $f_{\boldsymbol{\phi}}$ can be freely chosen to compensate for it.

**Computing the LML.** As mentioned in Section 5, when client $i$ employs a GP prior, $P_{\boldsymbol{\phi}}$, the negative log likelihood loss, and sets $\beta = m_i$, the quantity $\ln Z_{\beta}^C(P_{\boldsymbol{\phi}}, \mathcal{S}_i)$ defined in Corollary 3.4 corresponds to the LML of the GP. The closed-form formula for the LML is as follows:

$$
\begin{aligned}
\ln Z_{\beta}^C(P_{\boldsymbol{\phi}}, \mathcal{S}_i) = \ln \Pr[\mathbf{y}_i|\mathbf{X}_i, P_{\boldsymbol{\phi}}] = &-\frac{1}{2}(\mathbf{y}_i - \mathbf{m}_{\boldsymbol{\phi},i})^\top (\mathbf{K}_{\boldsymbol{\phi},i} + \sigma_{\boldsymbol{\phi}}^2 \mathbf{I})^{-1}(\mathbf{y}_i - \mathbf{m}_{\boldsymbol{\phi},i}) \\
&-\frac{1}{2}\ln|\mathbf{K}_{\boldsymbol{\phi},i} + \sigma_{\boldsymbol{\phi}}^2 \mathbf{I}| - \frac{m_i}{2}\ln(2\pi),
\end{aligned} \tag{33}
$$

where $|\cdot|$ denotes the determinant of a matrix. The vector $\mathbf{m}_{\boldsymbol{\phi},i} \in \mathbb{R}^{m_i}$ contains the output of $m_{\boldsymbol{\phi}}$ applied on each data sample and the matrix $\mathbf{K}_{\boldsymbol{\phi},i} \in \mathbb{R}^{m_i \times m_i}$ is the kernel matrix associated with the kernel function $k_{\boldsymbol{\phi}}$ applied to $\mathbf{X}_i$.

The determinant term in (33) is commonly viewed as a complexity penalty for the kernel (Rasmussen & Williams, 2005). However, recent findings in Rothfuss et al. (2021) suggest that this form of complexity regularization may be inadequate when dealing with expressive kernels that possess numerous hyperparameters, such as deep kernels. Furthermore, there is no complexity penalty imposed on the prior mean, which can lead to a higher risk of overfitting. In PAC-PFL, we address these limitations by using (9) as the loss function, which includes a KL divergence term between the hyper-posterior and the hyper-prior. The KL term penalizes hyper-posteriors that deviate significantly from the hyper-prior, thereby effectively regularizing both the mean and the kernel of the GP prior.

**Computing the predictive posterior.** Given a set of priors $P_{\boldsymbol{\phi}_1}, \cdots, P_{\boldsymbol{\phi}_k}$ trained by PAC-PFL, our objective is to compute the predictive posterior at a test point $\mathbf{x}_*$. We denote by $\hat{\mathcal{Q}}$ a uniform distribution over these priors, which serves as the SVGD approximation to the true optimal hyper-posterior, $\mathcal{Q}^*$. By introducing a categorical random variable $Z$ that is distributed uniformly over $1, \cdots, k$ and denotes which

of the priors $P_{\phi_1}, \cdots, P_{\phi_k}$ is used for making inference, the predictive posterior can be expressed as follows:

$$
\begin{aligned}
\Pr[y_*|\mathbf{x}_*, \mathcal{S}_i, \hat{\mathcal{Q}}] &= \sum_{\kappa=1}^{k} \Pr[y_*|\mathbf{x}_*, \mathcal{S}_i, \hat{\mathcal{Q}}, Z = \kappa] \Pr[Z = \kappa|\mathbf{x}_*, \mathcal{S}_i, \hat{\mathcal{Q}}] \\
&= \sum_{\kappa=1}^{k} \Pr[y_*|\mathbf{x}_*, \mathcal{S}_i, P_{\phi_\kappa}] \Pr[Z = \kappa|\mathbf{x}_*, \mathcal{S}_i, \hat{\mathcal{Q}}] \\
&= \sum_{\kappa=1}^{k} \Pr[y_*|\mathbf{x}_*, \mathcal{S}_i, P_{\phi_\kappa}] \Pr[\mathbf{x}_*, \mathcal{S}_i|P_{\phi_\kappa}] \frac{\Pr[Z = k|\hat{\mathcal{Q}}]}{\Pr[\mathbf{x}_*, \mathcal{S}_i|\hat{\mathcal{Q}}]} & (\textit{Bayes rule}) \\
&= \frac{1/k}{\Pr[\mathbf{x}_*, \mathcal{S}_i|\hat{\mathcal{Q}}]} \sum_{\kappa=1}^{k} \Pr[y_*|\mathbf{x}_*, \mathcal{S}_i, P_{\phi_\kappa}] \Pr[\mathbf{x}_*, \mathcal{S}_i|P_{\phi_\kappa}] & (\hat{\mathcal{Q}} \textit{ is uniform}).
\end{aligned} \tag{34}
$$

The first term in the summand of (34) is the predictive posterior distribution for client $i$ corresponding to the GP prior $P_{\phi_\kappa}(h) = \mathcal{GP}(h|m_{\phi_\kappa}, k_{\phi_\kappa})$, which is given by Rasmussen & Williams (2005) as:

$$
\Pr[y_*|\mathbf{x}_*, \mathcal{S}_i, P_{\phi_\kappa}] = \mathcal{N}(y_*|\mu_\kappa, \Sigma_\kappa), \tag{35a}
$$
$$
\mu_\kappa := m_{\phi_\kappa}(\mathbf{x}_*) + \mathbf{k}_{\phi_\kappa, i, *}^T (\mathbf{K}_{\phi_\kappa, i} + \sigma_{\phi_\kappa}^2 \mathbf{I})^{-1} (\mathbf{y}_i - \mathbf{m}_{\phi_\kappa, i}), \tag{35b}
$$
$$
\Sigma_\kappa := k_{\phi_\kappa}(\mathbf{x}_*, \mathbf{x}_*) - \mathbf{k}_{\phi_\kappa, i, *}^T (\mathbf{K}_{\phi_\kappa, i} + \sigma_{\phi_\kappa}^2 \mathbf{I})^{-1} \mathbf{k}_{\phi_\kappa, i, *} + \sigma_{\phi_\kappa}^2 \mathbf{I}, \tag{35c}
$$

where $\mathbf{k}_{\phi, i, *} \in \mathbb{R}^{m_i}$ is computed using the kernel function $k_\phi$ between $\mathbf{X}_i$ and $\mathbf{x}_*$.

Based on the assumption that $\mathbf{x}_*$ is independent of all samples in $\mathcal{S}_i$, we can expand the second term in (34) as follows:

$$
\begin{aligned}
\Pr[\mathbf{x}_*, \mathcal{S}_i|P_{\phi_\kappa}] &= \Pr[\mathbf{x}_*|P_{\phi_\kappa}] \Pr[\mathbf{X}_i|P_{\phi_\kappa}] \Pr[\mathbf{y}_i|\mathbf{X}_i, P_{\phi_\kappa}] \\
&= \Pr[\mathbf{x}_*] \Pr[\mathbf{X}_i] \Pr[\mathbf{y}_i|\mathbf{X}_i, P_{\phi_\kappa}] \\
&= \Pr[\mathbf{x}_*] \Pr[\mathbf{X}_i] \mathcal{N}(\mathbf{y}_i|m_{\phi_\kappa}(\mathbf{X}_i), k_{\phi_\kappa}(\mathbf{X}_i, \mathbf{X}_i) + \sigma_{\phi_\kappa}^2 \mathbf{I}),
\end{aligned} \tag{36}
$$

where the last line is the predictive distribution of a GP before conditioning on the observed data.

By merging (34)-(36), one obtains

$$
\Pr[y_*|\mathbf{x}_*, \mathcal{S}_i, \hat{\mathcal{Q}}] = \sum_{\kappa=1}^{k} \alpha_\kappa \mathcal{N}(y_*|\mu_\kappa, \Sigma_\kappa), \tag{37a}
$$
$$
\alpha_\kappa := \frac{\Pr[\mathbf{x}_*] \Pr[\mathbf{X}_i]}{k \Pr[\mathbf{x}_*, \mathcal{S}_i|\hat{\mathcal{Q}}]} \mathcal{N}(\mathbf{y}_i|m_{\phi_\kappa}(\mathbf{X}_i), k_{\phi_\kappa}(\mathbf{X}_i, \mathbf{X}_i) + \sigma_{\phi_\kappa}^2 \mathbf{I}) \in [0, 1] \tag{37b}
$$

It is straightforward to show that $\sum_{\kappa=1}^{k} \alpha_\kappa = 1$ in equation (37b). As a result, the predictive posterior in equation (37a) is a valid distribution over $y_*$.

### 8.3.3 Background on BNN

In this section, we introduce Bayesian Neural Networks (BNNs) used for classification tasks. Let $h_{\boldsymbol{\theta}} : R_{\mathbf{x}} \to R_y$ represent a neural network, with $\boldsymbol{\theta} \in \Theta$ denoting its parameters. Utilizing this mapping, we establish the conditional distribution as a Categorical distribution, derived as follows:

$$
\Pr\left[y|\boldsymbol{x}, \boldsymbol{\theta}\right] = \Pr\left[y\big|\textit{Categorical}\left(\textit{softmax}\left(h_{\boldsymbol{\theta}}(\boldsymbol{x})\right)\right)\right].
$$

**Computing the LML.** Unlike for GPs, the LML,

$$
\ln Z_\beta^C(P_\phi, \mathcal{S}_i) = \ln \mathbb{E}_{\boldsymbol{\theta} \sim P_\phi} e^{\frac{-\beta}{m_i} \sum_{\mathbf{z} \in \mathcal{S}_i} \ell(h_{\boldsymbol{\theta}}, \mathbf{z})},
$$

is intractable for BNNs. Instead, we use the following formula, as proposed by Rothfuss et al. (2021), to approximate the LML:

$$\ln \tilde{Z}_\beta^C(P_\phi, \mathcal{S}_i) = \ln LSE_{j=1}^L \left( e^{\frac{-\beta}{m_i} \sum_{\mathbf{z} \in \mathcal{S}_i} \ell(h_{\boldsymbol{\theta}_j}, \mathbf{z})} \right) - \ln L, \tag{38}$$

where $LSE$ is the LogSumExp function. This formula involves drawing $L$ samples $\boldsymbol{\theta}_1, \cdots, \boldsymbol{\theta}_L$ from $P_\phi$ to approximate the LML. We utilize this formula for LML approximation in the context of BNNs.

### 8.3.4 Background on SVGD

SVGD approximates a target probability distribution using a discrete uniform distribution over a set of designated samples called *particles* (Liu & Wang, 2016). Through an iterative process, the particles are updated by minimizing the KL divergence between the estimated and the target distributions in the reproducing kernel Hilbert space corresponding to a kernel function, $k_{SVGD}$. We utilize an RBF (Radial Basis Function) kernel with a heuristically chosen length scale, as described in Liu & Wang (2016). In our framework, we aim to obtain samples $P_{\phi_1}, \cdots, P_{\phi_k}$ from the target distribution $\mathcal{Q}^*$. To simplify the notation, we represent the particles as $\phi_1, \cdots, \phi_k$, where each particle $\phi_\kappa$ fully characterizes the corresponding prior, $P_{\phi_\kappa}$. Accordingly, we write $\mathcal{P}(\phi_\kappa)$ and $\mathcal{Q}^*(\phi_\kappa)$ instead of $\mathcal{P}(P_{\phi_\kappa})$ and $\mathcal{Q}^*(P_{\phi_\kappa})$.

Initially, the particles are sampled independently from the hyper-prior. Then, at each iteration, each particle $\phi_\kappa$ for $\kappa \in \{1, \cdots, k\}$ is updated according to the following rule:

$$\phi_\kappa \leftarrow \phi_\kappa + \frac{\eta}{k} \sum_{l=1}^k \left( k_{SVGD}(\phi_l, \phi_\kappa) \nabla_{\phi_l} \ln \mathcal{Q}^*(\phi_l) + \nabla_{\phi_l} k_{SVGD}(\phi_l, \phi_\kappa) \right), \tag{39}$$

where $\eta$ is the learning rate at the current iteration. By utilizing the formula for $\mathcal{Q}^*$ provided in Corollary 4.4 when $\beta = m_i$, we can derive the following expression for $\kappa \in \{1, \cdots, k\}$:

$$\nabla_{\phi_\kappa} \ln \mathcal{Q}^*(\phi_\kappa) = \nabla_{\phi_\kappa} \ln \mathcal{P}(\phi_\kappa) + \tau \sum_{i=1}^n \nabla_{\phi_\kappa} \ln Z_{m_i}^C(P_{\phi_\kappa}, \mathcal{S}_i), \tag{40}$$

where $\tau$ is defined in Corollary 4.4. By comparing (39) and (40), it becomes evident that the data of client $i$ is solely involved in the SVGD update through the term $\nabla_{\phi_\kappa} \ln Z_{m_i}^C(P_{\phi_\kappa}, \mathcal{S}_i)$ for each particle $\phi_\kappa$. As a result, the clients only need to transmit the gradient vector, $[\nabla_{\phi_1} \ln Z_{m_i}^C(P_{\phi_1}, \mathcal{S}_i), \cdots, \nabla_{\phi_k} \ln Z_{m_i}^C(P_{\phi_k}, \mathcal{S}_i)]^T$, or an approximation of it using a mini-batch approach, to the server. This observation is the intuition behind the sub-routine *Client_Update* in Algorithm 1 which we present in detail in the next subsection.

### 8.3.5 Table of parameters

We provide a comprehensive overview of the parameters relevant to our theoretical results and to our algorithm, along with their interrelationships in Table 4. To enhance clarity, the parameters are categorized into three groups, separated by horizontal lines. The first group describes the fundamental attributes of the FL problem, such as the number of clients, and is set externally. In the second group, we list the parameters central to our theoretical findings, elucidating their relations to other parameters. Lastly, the third group encapsulates the parameters used in implementing Algorithm 1, explaining their role in the practical application of our proposed methodology. Importantly, it should be noted that the constraints on the parameters have been deliberately set to ensure non-vacuous bounds, as discussed in Appendix 8.3.1.

### 8.3.6 Computational complexity

In this section, we discuss the computational complexity of a single iteration in training PAC-PFL, specifically when employing the log-likelihood loss and $\beta = m_i$, as outlined in the paper. At the client level, the computational complexity hinges on the calculation of the LML. For GPs, this complexity is $\mathcal{O}(km_i^3)$, where $k$ is the number of SVGD particles, and $m_i$ denotes the size of the training dataset for the specific client $i$. For BNNs, the complexity is $\mathcal{O}(kLm_i)$, where $L$ corresponds to the number of samples used to approximate the LML, as explained in Appendix 8.3.3.

| | Parameter | Description | Domain | Selection in experiments |
|---|---|---|---|---|
| problem setup | $n$ | number of clients | $\in \mathbb{N}$ | given in the dataset |
| | $n_2$ | number of clients with new samples | $\in \{0, \cdots, n\}$ | $n$ |
| | $\mathcal{T}$ | distribution of clients | - | unknown |
| | $m_i$ | number of samples for client $i$ | $\in \mathbb{N}$ | given in the dataset |
| | $\mathcal{D}_i$ | data generating distribution of client $i$ | $(\mathcal{D}_i, m_i) \sim \mathcal{T}$ | unknown |
| | $\mathcal{S}_i$ | dataset of client $i$ | $\sim \mathcal{D}_i^{m_i}$ | given in the dataset |
| | $\tilde{m}_i$ | number of new samples for client $i$ | $\in \{0, \cdots, m_i\}$ | given in the dataset |
| | $\tilde{\mathcal{S}}_i$ | new dataset of client $i$ | $\sim \mathcal{D}_i^{\tilde{m}_i}$ | given in the dataset |
| | $\mathcal{H}$ | model class | - | GP and BNN |
| client inference | $\ell$ | loss | $\in [a, b]$ s.t. $b - a < 8$ | negative log-likelihood |
| | $P$ | prior | over $\mathcal{H}$ | $\sim \mathcal{Q}^*$ |
| | $Q_i^*$ | optimal posterior for client $i$ | over $\mathcal{H}$ | Corollary 3.4 |
| | $\beta$ | temperature of $Q_i^*$ | $\in \mathbb{R}_+$ | $m_i$ (Appendix 8.3.1) |
| | $\delta$ | confidence level | $\in (0, 1]$ | see code |
| | $\epsilon_i$ | DP parameter | $\in \mathbb{R}_+$ | $\frac{2\beta\tau(b-a)}{m_i}$ (Lemma 4.5) |
| server inference | $\mathcal{P}$ | hyper-prior | - | $Gaussian\,(\mathbf{0}, \sigma_{\mathcal{P}}^2\,\mathbf{I})$ |
| | $\sigma_{\mathcal{P}}^2$ | variance of $\mathcal{P}$ | $\in \mathbb{R}_+$ | tuned by cross-validation |
| | $\mathcal{Q}^*$ | optimal hyper-posterior | - | Corollary 4.4 |
| | $\upsilon$ | constant in Theorem 4.2 | $\in \mathbb{R}_+$ | $10^{-4}$ |
| | $\lambda$ | constant in Theorem 4.2 | $\geq n_2 + \upsilon$ s.t. $\epsilon_i \leq \sqrt{2(b-a)}$ | tuned by cross-validation |
| | $\tau$ | temperature of $\mathcal{Q}^*$ | $\in \mathbb{R}_+$ | $\frac{\lambda}{\lambda + \beta n(n_2 + \upsilon)}$ (Corollary 4.4) |
| algorithm | $\eta$ | learning rate | $\in \mathbb{R}_+$ | tuned by cross-validation |
| | $k$ | number of SVGD particles | $\in \mathbb{N}$ | see code |
| | $T$ | number of iterations | $\in \mathbb{N}$ | see code |
| | $c$ | number of clients per iteration | $\in \{1, \cdots, n\}$ | see code |
| | $b$ | data batch size for each client | $\in \{1, \cdots, m_i\}$ | see code |
| | $L$ | number of samples for estimating the MLL of BNNs | $\in \mathbb{N}$ | see code |

Table 4: Summary of the employed notation and their relations. The parameter domains are set such that the bounds are non-vacuous. The parameters are categorized into four groups separated by horizontal lines: the first group describes the characteristics of the PFL problem. The second and third groups enumerate the notation used in our theoretical results. The final group encompasses the parameters in Algorithm 1.

At the server level, the computational complexity varies based on the type of hyper-prior used. When utilizing a hyper-prior with a diagonal covariance matrix, the complexity is $\mathcal{O}(ck + k^2)$, where $c$ represents the number of clients selected per iteration. If a hyper-prior with a full covariance matrix is used, the complexity increases to $\mathcal{O}(ck + k^3)$.

The training time (in seconds) of PAC-PFL and some baselines (where we optimize the computation time as much as possible without sacrificing the performance) for our classification experiments are provided in Table 5. As can be seen in the table, while the training time of PAC-PFL exceeds that of the simpler baselines, it remains within the same order of magnitude.

## 8.4 Role of DP

We adopt DP in two key ways. Firstly, we utilize DP to ensure that when the server samples a prior distribution from the hyper-posterior and releases it, this published prior distribution does not raise privacy

| ALGORITHM | EMNIST | FEMNIST (20) | FEMNIST (500) |
|---|---|---|---|
| FedAvg | 900 | 120 | 480 |
| MTL | 950 | 120 | 500 |
| MAML | 1100 | 540 | 605 |
| PAC-PFL | 1260 | 960 | 970 |

Table 5:   Training time (in seconds) of PAC-PFL and some baselines. The training time of PAC-PFL is slightly longer than that of more basic baselines but remains within the same order of magnitude.

concerns for the existing clients. This application is akin to typical scenarios in FL research where DP is used to protect the model shared by the server from revealing information about clients' data. Besides addressing privacy concerns, we use DP to establish a PAC bound at the client level. As illustrated in Figure 1, our setup involves prior distributions that depend on the data of existing clients. While most PAC bounds assume the prior distribution was selected before observing any data, we leverage results from Dziugaite & Roy (2018) to derive a PAC bound that holds when a data-dependent prior is obtained through a differentially private algorithm. To the best of our knowledge, DP has not been employed for this purpose in previous FL methods. Below, we discuss the role of DP in our ideal setup (Section 4) and our practical algorithm (Section 5).

**Inherent privacy of the optimal hyper-posterior**    In Section 4, we considered a specific family of hyper-posteriors, where sampling a prior from the hyper-posterior satisfies $\epsilon$-DP for a finite $\epsilon$. This assumption simplifies computations and enables us to use the closed-form posterior provided in Corollary 3.4. We then established the server-level upper bound in Theorem 4.2 for hyper-posteriors meeting the privacy criterion. Interestingly, the resulting upper bound does not contain the parameter $\epsilon$. This can be intuitively explained by the fact that the hyper-prior is chosen independently of the data, making the server-level scheme akin to typical PAC-Bayesian bounds that employ data-independent priors. Additional details can be found in the proof presented in Appendix 8.2. In summary, the $\epsilon$-DP assumption facilitates the computations by allowing us to use the optimal posterior formula without directly impacting the server-level bound.

By minimizing the server-level upper bound, we derived the closed-form formula for the optimal hyper-posterior in Corollary 4.4. Subsequently, we ensured that the optimal hyper-posterior satisfies the privacy assumption we started with. To achieve this, Lemma 4.5 determined the value of $\epsilon$ for which sampling the prior from the optimal hyper-posterior meets $\epsilon$-DP. As this $\epsilon$ is finite, it confirms that our optimal hyper-posterior belongs to the family of hyper-posteriors we initially considered, thus completing the derivations. While assuming a finite $\epsilon$ is sufficient for deriving the intended results, it is worth noting that the client-level bound becomes looser for larger values of $\epsilon$. Further discussion on providing non-vacuous bounds can be found in Appendix 8.3.1.

In the ideal setup, DP arises from the inherent randomness in sampling from the optimal hyper-posterior. As a result, DP is achieved without the need for externally injecting noise, which is a common practice in typical DP mechanisms. We refer to this property as the *inherent DP* of the optimal hyper-posterior.

**Loss of inherent DP due to SVGD**    As discussed in Section 5, since SVGD is a deterministic sampling algorithm, we forfeit the inherent privacy of $\mathcal{Q}^*$. Consequently, the client-level and server-level bounds no longer hold for the approximate hyper-posterior. Instead, we rely on the premise that if SVGD effectively approximates the optimal hyper-posterior, the empirical and true risks of the approximated and optimal hyper-posteriors should be closely aligned, suggesting the bound's validity.

**Differentially private PAC-PFL**    To reintroduce $\epsilon$-DP after SVGD approximation, Algorithm 1 can be modified, drawing inspiration from differentially private FedAvg (Geyer et al., 2017). In this modified version, the server clips the norm of the gradient sent by each client and introduces noise to the aggregated gradient. For simplicity, we will illustrate the algorithm using only one SVGD particle. A pseudo-code for this private version of PAC-PFL is provided in Algorithm 3.

---

**Algorithm 3** Differentially private PAC-PFL with 1 SVGD particle executed by the server

---

1: **Input:** privacy parameter $\epsilon$, gradient clipping norm $\gamma$, hyper-prior $\mathcal{P}$, parameter $\tau$, number of iterations $T$, number of clients per iteration $c$, mini-batch size $b$, learning rate $\eta$

2: Initialize prior $P_\phi \sim \mathcal{P}$ ▷ *Initialize*

3: **for** $t = 1$ to $T$ **do**

4:   Select a random subset $\mathcal{C}_t$ of $c$ clients

5:   **for** each selected client $i$ in $\mathcal{C}_t$ **in parallel do**

6:    $\boldsymbol{G}_i \leftarrow Client\_Update(b, \boldsymbol{\phi})$

7:    Clip gradient norm: $\hat{\boldsymbol{G}}_i \leftarrow \boldsymbol{G}_i / \max\left(1, \frac{\|\boldsymbol{G}_i\|_2}{\gamma}\right)$    ▷ *Collect clipped client updates*

8:   Sample noise: $\boldsymbol{\nu} \sim Lap(\boldsymbol{0}, \frac{T\gamma}{\epsilon c}\boldsymbol{I}_{d_\phi})$

9:   $\hat{\boldsymbol{G}} \leftarrow \frac{1}{c}\sum_{i\in\mathcal{C}_t}\hat{\boldsymbol{G}}_i + \boldsymbol{\nu}$    ▷ *Aggregate gradients and inject noise*

10:   **for** $\kappa = 1$ to $k$ **do**

11:    $\nabla_{\boldsymbol{\phi}_\kappa}\ln\mathcal{Q}^*(\boldsymbol{\phi}_\kappa) \leftarrow \nabla_{\boldsymbol{\phi}_\kappa}\ln\mathcal{P}(\boldsymbol{\phi}_\kappa) + \tau\,\hat{\boldsymbol{G}}_{\kappa*}^T$    ▷ *$\hat{\boldsymbol{G}}_{\kappa*}$ is the $\kappa$-th row of $\hat{\boldsymbol{G}}$*

12:   **for** $\kappa = 1$ to $k$ **do**

13:    $\boldsymbol{\phi}_\kappa \leftarrow \boldsymbol{\phi}_\kappa + \frac{\eta}{k}\sum_{l=1}^{k}\left(k_{SVGD}(\boldsymbol{\phi}_l, \boldsymbol{\phi}_\kappa)\nabla_{\boldsymbol{\phi}_l}\ln\mathcal{Q}^*(\boldsymbol{\phi}_l) + \nabla_{\boldsymbol{\phi}_l}k_{SVGD}(\boldsymbol{\phi}_l, \boldsymbol{\phi}_\kappa)\right)$   ▷ *SVGD update*

14: **return** $P_{\boldsymbol{\phi}_1}$    ▷ *Differentially-private SVGD approximation of $\mathcal{Q}^*$*

---

As we are considering a single particle in Algorithm 3, the *Client_Update* function returns a vector $\boldsymbol{g}_i$ instead of a matrix $\boldsymbol{G}_i$, as is the case when using multiple SVGD particles. In Algorithm 3, the notation $Lap(\boldsymbol{0}, \frac{T\gamma}{\epsilon c}\boldsymbol{I}d\phi)$ represents a multivariate Laplace probability distribution. In this context, $\boldsymbol{0}$ denotes a zero mean vector, and $\frac{T\gamma}{\epsilon c}\boldsymbol{I}d\phi$ specifies the scale parameter applied across all dimensions. Both the particle, $\boldsymbol{\phi}$, and the noise vector, $\nu$, have dimension $d_\phi$.

The noise scale in Algorithm 3 grows linearly with the number of iterations, $T$. This can introduce significant noise into the gradients, potentially impairing both convergence and the overall accuracy of the algorithm. This challenge is intrinsic to differentially private gradient descent and is not unique to our model or FL in general (Bagdasaryan et al., 2019). Investigating alternative strategies to achieve $\epsilon$-DP remains a promising avenue for future research.

We assess the performance of Algorithm 3 using the Polynomial dataset, introduced in Appendix 8.5. Specifically, we use 120 existing clients, each with 10 samples. We vary the privacy parameter $\epsilon$ and examine its impact on the algorithm's performance, measured by the average RSMSE metric. Figure 4 illustrates the RSMSE across different values of $\epsilon$. Additionally, we plot the RSMSE of the non-private algorithm, Algorithm 1, in red for reference. As anticipated, the model's performance improves as $\epsilon$ increases, signifying a lower level of privacy and consequently reduced noise requirements.

### 8.5 Datasets

### 8.5.1 PV dataset

The transition to renewable energy sources, such as roof-top photovoltaic (PV) panels, is crucial to address energy challenges; however, the intermittency of solar energy remains a challenge that requires accurate prediction of solar panel power output. Predicting the time series of PV panel power outputs requires either specific measurements or large amounts of data. A potential solution is to design a collaborative methodology among multiple PV datasets collected at nearby locations for predictive modeling.

We work with an hourly simulated dataset of PV generation time series from rooftop panels in Lausanne, Switzerland. Our objective is to predict the next-hour PV generation, utilizing 15 features encompassing auto-regressors and weather data. We investigate four scenarios: *PV-S (150)*, *PV-S (610)*, *PV-EW (150)*, and *PV-EW (610)*. In the *PV-S* variants, all houses face south, while the *PV-EW* variants involve houses oriented either east or west. The *PV-EW* scenarios, being bimodal and highly heterogeneous, present

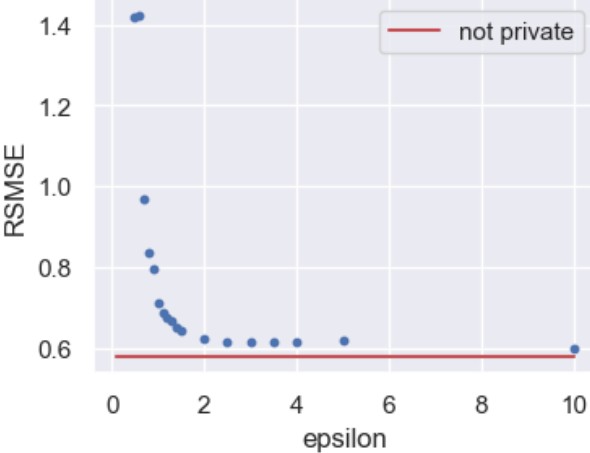

Figure 4: Performance evaluation of differentially private PAC-PFL (Algorithm 3) on the Polynomial dataset introduced in Appendix 8.5. The RSMSE metric is plotted against varying values for the differential privacy parameter, $\epsilon$. Lower $\epsilon$ values correspond to lower privacy levels. The RSMSE of non-private PAC-PFL (Algorithm 1) is shown in red for reference. As expected, performance improves as the privacy level decreases due to lower noise injection.

greater modeling challenges. To assess the impact of dataset sizes (denoted as **m**), we consider two settings: clients with either 150 or 610 training samples, corresponding to two-week and two-month data windows, respectively. The number in each dataset's name indicates the training sample size.

**Data generation.**  To generate simulated datasets for the PV-S and PV-EW experiments, we first obtained a real dataset of hourly solar radiation and meteorological measurements in Lausanne from the *Photovoltaic Geographical Information System* (PVGIS) online tool (PVG). We then used the *pvlib* Python library (Holmgren et al., 2022) to simulate the PV power output based on these measurements. To simulate the PV panels in different houses, we sampled meteorological data and installation specifications from normal distributions with specified means and standard deviations. For example, in the PV-EW experiment, the azimuths of the PV panels were sampled from a bimodal normal distribution of $0.5\big(\mathcal{N}(90°, 15°) + \mathcal{N}(270°, 15°)\big)$, where $\mathcal{N}$ denotes the normal distribution. Figure 5 illustrates the power output profiles of 24 houses in the PV-EW experiment over five days. This figure demonstrates that the curves have noticeable differences; however, they show similar trends. As expected, the houses facing east or west are more similar to one another, while the differences among different sub-populations are higher. For instance, the peak production of the houses facing the east occurs in the morning, whereas the peak for those facing the west occurs in the afternoon.

To ensure that each client's dataset is identically distributed, despite the variations in meteorological patterns throughout the year, we filter the dataset down to the months of June and July. We employ two distinct training datasets for our analysis. The first dataset comprises the initial two weeks of June 2018, which provides a total of 150 samples for each client. The second dataset encompasses the data from both June and July 2018, resulting in 610 training samples per client. For all experiments, the test dataset consists of the data from June and July 2019. Furthermore, we exclude nighttime data points when the PV generation is zero. To normalize the data, we standardize each house's features and output, setting the mean to zero and the standard deviation to one.

**Features.**  The dataset contains various meteorological features, including solar beam and diffuse irradiances, temperature, wind speed, solar altitude, time, and date. Solar beam and diffuse irradiances are particularly valuable in predicting PV generation, but their measurement can be costly. To reflect a realistic scenario, we assume that all households have access to the irradiance data recorded at a specific location in the city, such as a weather station. However, they are unaware of the specific irradiance values at their own houses, which may differ from the weather station. The temperature and wind speed may vary slightly

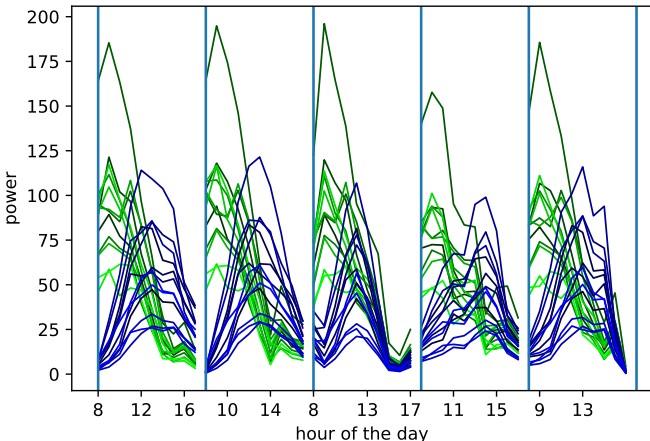

Figure 5: Power output profile of 24 houses in the PV-EW experiment over five days in June 2018, where each line represents the PV generation of one house. Green and blue curves correspond to houses facing the east and the west respectively. Although the curves have noticeable differences, there are consistent trends present in the data.

among different houses in the dataset. Additionally, some houses may experience intermittent shadows caused by nearby trees or buildings, which can appear and disappear at certain times of the day. These shadows are considered as noise, and no recorded feature in the dataset provides explicit information about their occurrence or characteristics.

In addition to the meteorological features, we also incorporate autoregressors in our analysis. Autoregressors are advantageous in time-series prediction tasks and aim to capture the temporal dependencies in the data. To determine the autoregressors to include in our analysis, we utilize the partial autocorrelation function (PACF). The PACF measures the correlation between a time series and its lagged values while controlling for the correlations with all shorter lags. We select autoregressors with the highest values of the PACF among the lagged values up to two weeks ago. By focusing on these highly correlated autoregressors, we aim to capture the most relevant information from the past time steps.

### 8.5.2 Polynomial dataset

We examine a bimodal dataset where the data for each client is generated by sampling a function from one of two GP priors. Each GP prior is characterized by a polynomial mean function of order 7 and an SE kernel function. The two modes have distinct polynomial means and length scales associated with the SE kernel. Additionally, Gaussian noise is introduced into the generated data to account for measurement errors and other sources of variability. Figure 6 illustrates the dataset for a total of 24 clients, with 12 clients belonging to the first mode and another 12 clients belonging to the second mode, where each client has 10 training samples.

### 8.5.3 FEMNIST and EMNIST Datasets

The FEMNIST (Federated Extended MNIST) (Caldas et al., 2019) and EMNIST (Extended MNIST) (Cohen et al., 2017) datasets consist of $28 \times 28$ images of hand-written English characters. Both datasets are used for handwritten character recognition. Specifically, the FEMNIST dataset is employed in a 10-way classification task, focusing on the classification of digits. Meanwhile, the EMNIST dataset is utilized for a more comprehensive 62-way classification task. This broader task encompasses the recognition of both lower case and upper case English alphabet letters, in addition to digits.

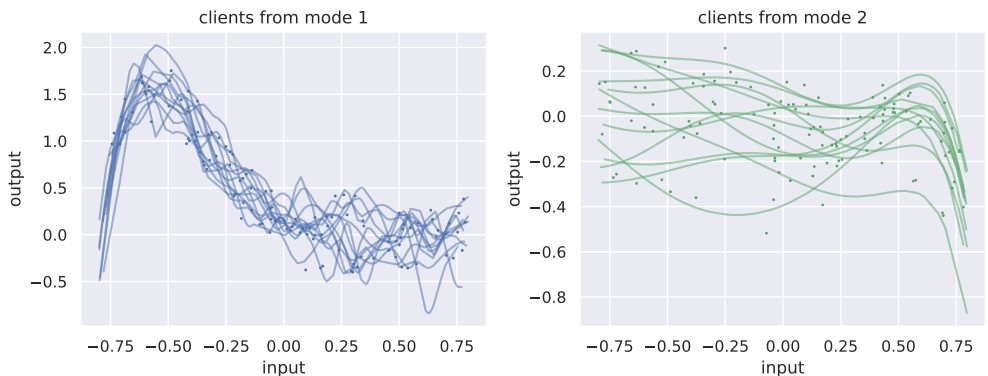

Figure 6: Polynomial dataset for 24 clients, where 12 clients belong to the first and the rest belong to the second mode. The solid lines represent the unknown true data-generating distribution for each client. Each client has a training dataset with 10 samples. The solid lines in the figure represent the true data-generating distribution, which is unknown to us. The points on the graph represent the noisy samples in each client's training dataset.

FEMNIST identifies the writer for each character, allowing a natural partitioning strategy by assigning all images from the same writer to a single client (Caldas et al., 2019). This dataset is heterogeneous due to the inherent diversity in handwriting styles, resulting in skewness within the feature distribution (Tan et al., 2021). For our analysis, we subsample 40 clients and explore two scenarios: *FEMNIST (*20*)*, where each client is allocated 20 samples and *FEMNIST (*500*)*, where each client utilizes all available samples, approximately 500 per client.

Following (Wang et al., 2020; Marfoq et al., 2021), we subsample 10% of the EMNIST dataset that amounts to 81425 total samples. We distribute samples with the same label across 80 clients according to a symmetric Dirichlet distribution with parameter $\alpha = 0.4$. This approach introduces a skew in the label distribution (Tan et al., 2021), thereby synthesizing heterogeneity in the dataset.

### 8.6 Experiments details

### 8.6.1 Calibration error

Calibration error (CE) is an evaluation metric relevant to probabilistic models. It is based on the premise that when provided with a test input $\mathbf{x}_j$, the model produces a probability distribution $\hat{p}(y_j|\mathbf{x}_j)$ over the predicted target $y_j$. CE measures the disparity between the predicted confidence intervals and the actual proportions of test data falling within those intervals (Kuleshov et al., 2018). CE is a nonnegative measure and a smaller CE value is more desirable.

**Calibration error for regression.** In defining CE for regression, we follow the formula proposed by (Rothfuss et al., 2021). Denote a predictor's cumulative density function (CDF) as $\hat{F}(y_j|\mathbf{x}_j) = \int_{-\infty}^{y_j} \hat{p}(y|\mathbf{x}_j)dy$. Given a dataset $\mathcal{S} = \{(\mathbf{x}_j, y_j)\}_{j=1}^{m}$ with $m$ samples, we compute the corresponding empirical frequency for confidence levels $0 \leq q_1 \leq \cdots \leq q_H \leq 1$ as follows:

$$\hat{q}_h := \frac{1}{m}\left|\left\{y_j \,\middle|\, \hat{F}(y_j|\mathbf{x}_j) \leq q_h, \quad j = 1, \cdots, m\right\}\right|,$$

for $h = 1, \cdots, H$.

If the predictions are well-calibrated, we expect that $\hat{q}_h \to q_h$ as $m \to \infty$. We adopt the CE definition proposed by Rothfuss et al. (2021), which formulates CE as a function of residuals $\hat{q}_h - q_h$:

$$CE := \frac{1}{H}\sum_{h=1}^{H}|\hat{q}_h - q_h|. \tag{41}$$

In our experiments, we evaluate (41) by employing $H = 20$ equally spaced confidence levels ranging from 0 to 1.

**Calibration error for classification.**   Given a test input $\mathbf{x}_j$, a probabilistic classifier outputs a categorical probability distribution, $\hat{p}(y_j = k|\mathbf{x}_j)$, where $k = 1, \cdots, C$, with $C$ representing the number of classes. The classifier's prediction is the class label with the highest probability, denoted as $\hat{y}_j = \arg\max_k \hat{p}(y_j = k|\mathbf{x}_j)$. Correspondingly, we define the classifier's confidence in the prediction for the input $\mathbf{x}_j$ as $\hat{p}_j \coloneqq \hat{p}(y_j = \hat{y}_j|\mathbf{x}_j)$.

For a well-calibrated classifier, the confidence is expected to align with the probability of correct classification. For example, with 100 predictions, each having a confidence of 0.8, one would anticipate approximately 80 correct classifications (Guo et al., 2017). In an empirical comparison of calibration and accuracy on a dataset $\mathcal{S} = \big\{(\mathbf{x}_j, y_j)\big\}_{j=1}^m$, following the methodology of Guo et al. (2017), data samples are grouped into $H = 20$ intervals, each of length $1/H$, based on their prediction confidence. Let $B_h \coloneqq \big\{j|\hat{p}_j \in (\frac{h-1}{H}, \frac{h}{H}]\big\}$ for $h = 1, \cdots, H$ be the set of indices of points in $\mathcal{S}$ whose prediction confidence falls within interval $h$. The accuracy and average confidence of points in $B_h$ are defined as follows:

$$acc(B_h) \coloneqq \frac{1}{|B_h|} \sum_{j \in B_h} \mathbf{1}(\hat{y}_j = y_j),$$

$$conf(B_h) \coloneqq \frac{1}{|B_h|} \sum_{j \in B_h} \hat{p}_j.$$

To compare the accuracy and average confidence across all intervals, Guo et al. (2017) calculates CE as follows:

$$CE \coloneqq \sum_{h=1}^H \frac{|B_h|}{m} |acc(B_h) - conf(B_h)|. \tag{42}$$

This formula measures the deviation between accuracy and prediction confidence level, serving as a metric for assessing the calibration of classification tasks.

### 8.6.2   Regression experiments details

**Baselines details.**   In the Vanilla approach, the GP hyperparameters for each client are tuned by maximizing the LML of that specific client, without using FL. For pooled GP, we use inducing points to handle computational issues due to the large number of samples involved in this approach. pFedGP optimizes the average LML across clients to obtain the deep kernel hyper-parameters. We adapted pFedGP, originally designed specifically for classification, to suit our regression task. With this adaptation, pFedGP is a special case of our algorithm when using a single prior in SVGD and a very wide hyper-prior. We try pFedGP with a zero GP mean, as originally done in Achituve et al. (2021), and a NN mean and do not use inducing points. We apply pFedBayes (Zhang et al., 2022) only to the FEMNIST data as the authors only consider classification tasks in their experiments.

**Hyper-parameter tuning.**   We perform hyper-parameter tuning for all baselines and our method using 5-fold cross-validation. For all neural networks, we explore structures with the same number of neurons per layer. The number of neurons per layer can take values of $2^n$ for $n \in 1, \cdots, 6$, and we consider 2 or 4 hidden layers. For PAC-PFL, we employ 4 SVGD particles and set $k = 4$. The parameter $\beta$ is set to the number of samples for each client, $\beta = m_i$. To determine the value of $\tau$, we search for values greater than $1/(1 + n\beta)$, ensuring that $\lambda > n_2 + \upsilon$ and thus, satisfying the assumption outlined in Theorem 4.2.

The employed hyper-prior is a multivariate Gaussian distribution with a diagonal covariance matrix. It is a distribution over the weights and biases of the mean and kernel neural networks, as well as the noise standard deviation of the likelihood. In all PV experiments, we set the hyper-prior mean for the neural network weights and biases to 0 and the hyper-prior mean for the noise standard deviation to 0.4. These choices help prevent overfitting.

Table 6: Comparison of probabilistic (⬛) and non-probabilistic (⬛) FL approaches along with probabilistic non-federated baselines (⬛) on regression tasks. Average RSMSE for all baselines (⬛, ⬛, ⬛) and CE for probabilistic approaches (⬛, ⬛) over 5 trials, where ± captures a 95% confidence interval are reported.

| | Dataset | PV-S (150) | | PV-EW (150) | | PV-S (610) | | PV-EW (610) | | Polynomial (10) | |
|---|---|---|---|---|---|---|---|---|---|---|---|
| | Metric | RSMSE | CE | RSMSE | CE | RSMSE | CE | RSMSE | CE | RSMSE | CE |
| Existing clients | PAC-PFL | 0.43 ± 0.02 | 0.07 ± 0.00 | 0.41 ± 0.01 | 0.04 ± 0.00 | 0.42 ± 0.01 | 0.07 ± 0.00 | 0.40 ± 0.01 | 0.04 ± 0.00 | 0.58 ± 0.05 | 0.14 ± 0.02 |
| | pFedGP | 0.48 ± 0.04 | 0.07 ± 0.02 | 0.45 ± 0.01 | 0.06 ± 0.01 | 0.51 ± 0.03 | 0.07 ± 0.00 | 0.46 ± 0.01 | 0.05 ± 0.01 | 0.80 ± 0.15 | 0.12 ± 0.01 |
| | MTL | 0.45 ± 0.00 | - | 0.43 ± 0.00 | - | 0.44 ± 0.01 | - | 0.41 ± 0.00 | - | 0.85 ± 0.15 | - |
| | MAML | 0.57 ± 0.02 | - | 0.52 ± 0.03 | - | 0.58 ± 0.02 | - | 0.52 ± 0.02 | - | 0.98 ± 0.13 | - |
| | Vanilla | 0.68 ± 0.02 | 0.12 ± 0.01 | 0.63 ± 0.03 | 0.12 ± 0.01 | 0.49 ± 0.04 | 0.04 ± 0.00 | 0.46 ± 0.02 | 0.03 ± 0.00 | 0.73 ± 0.07 | 0.18 ± 0.01 |
| | Pooled | 0.48 ± 0.03 | 0.26 ± 0.00 | 0.47 ± 0.05 | 0.26 ± 0.01 | 0.49 ± 0.02 | 0.26 ± 0.01 | 0.46 ± 0.03 | 0.27 ± 0.01 | 2.33 ± 0.55 | 0.33 ± 0.04 |
| New clients | PAC-PFL | 0.43 ± 0.02 | 0.07 ± 0.00 | 0.42 ± 0.01 | 0.04 ± 0.00 | 0.43 ± 0.01 | 0.07 ± 0.00 | 0.43 ± 0.01 | 0.04 ± 0.00 | 0.65 ± 0.02 | 0.16 ± 0.01 |
| | pFedGP | 0.46 ± 0.04 | 0.07 ± 0.02 | 0.45 ± 0.02 | 0.06 ± 0.01 | 0.51 ± 0.05 | 0.06 ± 0.01 | 0.46 ± 0.01 | 0.06 ± 0.01 | 0.87 ± 0.41 | 0.15 ± 0.02 |
| | MTL | 0.44 ± 0.00 | - | 0.42 ± 0.00 | - | 0.45 ± 0.00 | - | 0.43 ± 0.00 | - | 0.88 ± 0.12 | - |
| | MAML | 0.55 ± 0.03 | - | 0.52 ± 0.01 | - | 0.55 ± 0.01 | - | 0.53 ± 0.02 | - | 1.11 ± 0.28 | - |
| | Vanilla | 0.69 ± 0.02 | 0.12 ± 0.01 | 0.63 ± 0.05 | 0.12 ± 0.01 | 0.69 ± 0.02 | 0.12 ± 0.01 | 0.63 ± 0.52 | 0.12 ± 0.01 | 0.76 ± 0.05 | 0.24 ± 0.03 |
| | Pooled | 0.48 ± 0.03 | 0.26 ± 0.00 | 0.47 ± 0.05 | 0.26 ± 0.01 | 0.48 ± 0.02 | 0.27 ± 0.00 | 0.46 ± 0.03 | 0.27 ± 0.01 | 2.32 ± 0.37 | 0.36 ± 0.02 |

**Results.** In this section, we evaluate our algorithm and the introduced baselines based on the RSMSE and CE error metrics for existing and new clients. For a given method, we calculate the RSMSE and CE, where applicable, for each existing and new client using a fixed random seed. Then, we compute the average RSMSE and average CE metrics across the existing and new client groups. We repeat this analysis five times using different random seeds and calculate the 95% confidence interval for the sample mean of average RSMSE and average CE.

The results are presented in Table 6. In each column, the number after the dataset name represents the number of training samples per client. As shown in Table 6, our method outperforms other approaches in the majority of cases in terms of RSMSE for both existing and new clients. Furthermore, we observe that our method's performance improves with an increase in the number of training samples per client. Concerning CE, PAC-PFL consistently exhibits low CE values across all scenarios, establishing itself as the top-performing or closely competitive method. This observation aligns with findings in Rothfuss et al. (2021), where the authors noted that CE improvement is notable when the meta-learning tasks exhibit greater similarity. In our experiments, given the high client heterogeneity present in both PV-EW and PV-S, the improvements in CE are comparatively modest.

We believe there are three reasons why PAC-PFL outperforms other baselines. First, by learning the prior with FL and treating posterior inference as personalization, PAC-PFL exhibits a high level of adaptability to individual patterns. Second, PAC-PFL's ability to learn multiple priors enables it to effectively model clients with heterogeneous data distributions. Finally, the regularization of the hyper-posterior towards the hyper-prior in PAC-PFL helps prevent overfitting and allows for learning more complex prior means even with limited training data. For example, when selecting the GP prior mean structure through cross-validation, it is observed that the best GP mean for pFedGP is a linear function, while for PAC-PFL it is a 2-layer neural network with 32 neurons per layer. The use of a more complex prior mean in PAC-PFL allows for the potential capture of more intricate patterns in the data.

### 8.6.3 Classification experiments details

**Baselines implementations.** We implemented pFedGP Achituve et al. (2021) using the code provided by the authors without utilizing inducing points. Since the source code for pFedBayes has not been publicly released, we rely on an unofficial implementation[6]. Our implementations of FedAvg, MTL, and MAML are based on the code provided in Xie et al. (2023).

**Hyper-parameter tuning.** Hyper-parameter tuning for all baselines and our method is conducted using the cross-validation approach. In the case of PAC-PFL, we select $\beta$ and $\tau$ following the same procedure as employed for the PV dataset. We use 3 SVGD particles and a multivariate Gaussian hyper-prior with a diagonal covariance matrix.

---

[6]https://github.com/AllenBeau/pFedBayes

