# OpenReview forum: "Personalized Federated Learning of Probabilistic Models: A PAC-Bayesian Approach"
_TMLR — Accepted by TMLR_

### Review · Reviewer_xC1T · 2024-09-27

**Summary Of Contributions:**

This paper presents PAC - PFL, a novel algorithm addressing challenges in federated learning. It learns a shared hyper-posterior federatively for personalized inference, mitigates overfitting, and provides generalization bounds for new clients. Experiments show its effectiveness in accurate and well-calibrated predictions.

**Audience:**

Yes

**Claims And Evidence:**

Yes

**Requested Changes:**

Provide more motivation / real applications of PFL.

**Strengths And Weaknesses:**

Strengths:
1. The propose approach can handle highly heterogeneous clients by formulating a hyper-posterior, enhancing the flexibility to capture diverse patterns compared to methods with a single shared prior. It adapts well to individual client patterns by treating posterior inference as personalization after learning the prior with FL.
2. Theoretical results are established, providing guarantees on the performance and generalization ability of the model.
3. Experimental results show the proposed approach outperforms various baselines.

Weaknesses:
1. It would be better to provide more motivation / real applications of PFL.

---

> ### Author Response · Authors · 2024-12-06
> **Response to reviewer's comments**
>
> We are grateful for the reviewer’s constructive feedback.
> Following your comment, we have extended motivation for PFL in the introduction section of the updated version of the paper. The updated paragraph is as follows:
>
> “A key challenge in FL is the heterogeneity of clients’ datasets, which violates the i.i.d. assumption required for training the global model (Kairouz et al., 2021; Li et al., 2020a). This often leads to convergence difficulties or suboptimal performance for the global model (Li et al., 2020b). To address this issue, Personalized FL (PFL) introduces a personalization step to adapt the global model to the specific data of individual clients. This step is critical in many real-world federated datasets, as they typically involve heterogeneous clients (Wen et al., 2022). The growing impact of PFL has been highlighted in diverse applications, including image classification, regression, text analysis, and recommendation systems. (Chen et al., 2024).”
>
> We hope this update addresses your concern.

---

> > ### Comment · Reviewer_xC1T · 2024-12-09
> >
> > I thank the authors for their response, and acknowledge the changes made.

---

### Review · Reviewer_bAQB · 2024-10-19

**Summary Of Contributions:**

This paper studies personalized federated learning. The PAC-PFL method is introduced, where a shared (hyper)-posterior is learned and each client determines their posterior. The benefit of this approach is that there is no regularization towards a shared model, which enhance personalization. Furthermore, it can easily adapt to include new information. The authors claim that PAC-PFL is sota across all the datasets they evaluate.

**Audience:**

Yes

**Claims And Evidence:**

Yes

**Requested Changes:**

Would it be possible to re-work this paper to improve presentation? There is a lot of jargon. Even with an understanding of what all these words mean, it isn't very clear. prior of what? posterior of what? What makes something a "hyper"-posterior?

The notation is very cumbersome, is there anyway this can be improved?

Add some suitable parenthesis to Eq. 3

Do the results depend on the degree of heterogeneity?

**Strengths And Weaknesses:**

**Strengths**
* The experimental section seems strong, though I am not aware of the latest literature. (Given the claims please make sure the section is as thorough as possible.)
* Table 1 is a nice summary

**Weaknesses**
* Notation is hard to follow. I had difficulty reading this paper. Maybe more figures would help?
* Presentation is poor.

---

> ### Author Response · Authors · 2024-12-06
> **Response to reviewer's comments**
>
> We appreciate the reviewer’s insightful feedback and have addressed each comment in the order it was presented. Furthermore, we made additional refinements to enhance the paper’s quality according to the comments.
>
> **1. Improving the paper presentation and clarifying the jargon**
>
> Following your comment and feedback from reviewer J4a8, we have made the following changes to the updated version of the paper:
>
> * Added Definition 4.1 for defining the hyper-prior and hyper-posterior distributions.
>
> * Included further explanation about the prior, posterior, hyper-prior, and hyper-posterior distributions in the introduction.
>
> * Updated Figure 1, to better illustrate the role of different distributions involved in our framework.
>
> We believe this addition provides a more comprehensive understanding of these key concepts.
>
> **2. Improving the notation**
>
> The notation used in our paper is consistent with existing PAC-Bayesian meta-learning approaches, such as [1]. To facilitate understanding, we provided Table 4 in the submitted paper, which summarizes all employed notations for easy reference. We hope this clarification addresses your concern.
>
> [1] Jonas Rothfuss, Vincent Fortuin, Martin Josifoski, and Andreas Krause. PACOH: Bayes-optimal meta-learning with PAC-guarantees. In Proceedings of the 38th International Conference on Machine Learning
>
> **3. Add some suitable parenthesis to Eq. 3**
>
> We have fixed this issue in the updated version of the paper. Thank you for your suggestion.
>
> **4. Do the results depend on the degree of heterogeneity?**
>
> Our generalization bounds, derived using PAC-Bayes principles, do not explicitly depend on the degree of heterogeneity. This independence is a fundamental property of PAC-Bayes bounds, which are designed to provide guarantees that hold regardless of the heterogeneity level.
>
> In practice, higher heterogeneity increases the challenge of personalization, as it requires further tailoring of the trained model to individual clients' data. This challenge is not unique to our approach but affects all personalized federated learning algorithms. Despite this, we have demonstrated that our algorithm consistently outperforms baseline methods, even under highly heterogeneous scenarios, as shown in the PV-EW dataset experiments (Figure 2).

---

> > ### Comment · Reviewer_bAQB · 2025-01-03
> > **Response to Comment**
> >
> > Thank you for your response, and your revision to the paper. I think my comments have been addressed sufficiently.

---

### Review · Reviewer_J4a8 · 2024-11-29

**Summary Of Contributions:**

The work considers the problem of PFL (personalized federated learning) in a heterogenous settings. The proposed approach aims to address the following issues in PFL: a) lack of uncertainty quantification (which is address by using probabilistic models) b) inability to capture heterogeneity (addressed using a notion of "hyper-priors") c) performance degradation when client datasets are small  d) enabling inclusion of new datasets. The central idea is to adapt ideas from PAC-Bayesian meta-learning to PFL settings.

**Audience:**

Yes

**Claims And Evidence:**

No

**Requested Changes:**

1. Firstly, a clear demarkation needs to be done between existing work and the new approach being proposed. What is the net new of the work, this is not clear.
2. The experimental needs to demonstrate the issues discussed in the introduction -- particularly the heterogeneity, small client performance degradation etc. It needs to be a lot more extensive, in-lieu of theoretical results.
3. Define hyper-prior and hyper-posterior formally (perhaps within a definition environment)
4. Address the scalability blurb in the related work -- It is written in a hand-wavy fashion in my opinion.

**Strengths And Weaknesses:**

Strengths:

1. The paper does a good job describing the key challenges in the PFL domain.
2. The key idea to adapt meta-learning to PFL is interesting.

Weaknesses:

1. From a theoretical point of view, I think there is a lack of real novelty. Instead the paper heavily draws from prior art and the novelty is incremental at best. I am not an expert in the area, but it would be greatly helpful if you could clearly articulate the key challenges involved in adapting meta-learning ideas to the PFL domain. Separate out the new techniques from old techniques.
2. While the premise is quite interesting, the claims in the introduction are not substantiated to my understanding. For example, how are you demonstrating that extreme heterogeneity is tackled better than prior art? Further, the datasets are quite tiny, how do the techniques scale with larger datasets. Even so, there are only 3 datasets on which the approach is evaluated and only on one dataset the approach seems to beat meta learning approaches (which is not surprising the approach is borrowed from it).
3. Scalability is explicitly called out when discussing the work by Corinzia & Buhmann (2019). How is PAC-PFL better than that? It should be noted that while the primary concern seems to be "sequential" nature of fine-tuning models at the client, this can effectively be done in parallel, so I am not sure what the authors mean by "scalability".

---

> ### Author Response · Authors · 2024-12-06
> **Response to reviewer's comments - Part 1**
>
> Thank you for your thoughtful and detailed comments. We have responded to your feedback point by point and implemented improvements to the paper accordingly.
>
> **1. From a theoretical point of view, I think there is a lack of real novelty. Instead, the paper heavily draws from prior art and the novelty is incremental at best. Clearly articulate the key challenges involved in adapting meta-learning ideas to the PFL domain. Separate out the new techniques from old techniques.**
>
> While it is true that we leverage previous results, we believe our technical novelties are not only incremental. To better highlight this in the paper, we have added the paragraph “Novelties” in Section 2 of the updated paper. Below, we provide this paragraph for your convenience.
>
> “We propose PAC-PFL, a probabilistic PFL framework that addresses challenges (c1)-(c4) while introducing two significant advancements over existing methods. First, PAC-PFL improves upon Bayesian and PAC-Bayesian PFL approaches by offering a theoretically rigorous and data-efficient inference pipeline that eliminates the need for data splitting. This is achieved through PAC-Bayesian methods allowing for data-dependent priors that are differentially private (Dziugaite & Roy, 2018), along with an analysis of the proposed method’s privacy properties. Second, unlike PAC-Bayesian meta-learning approaches that primarily benefit new clients (tasks), PAC-PFL incentivizes existing clients to collaborate in the framework by learning a hyper-posterior distribution tailored for their data. To achieve this, we define a novel loss function that evaluates the hyper-posterior based on the accuracy of models derived via the pipeline in Figure 1 for existing clients. We then derive a generalization bound for this loss function, focusing on unseen data from existing clients rather than new clients, as in meta-learning. With these advancements, PAC-PFL provides a principled and compelling solution for addressing existing challenges in PFL.”
>
> We hope this addition better points out our technical contributions.
>
> **2. The claims in the introduction are not substantiated to my understanding. For example, how are you demonstrating that extreme heterogeneity is tackled better than prior art?  The experiment needs to demonstrate the issues discussed in the introduction.**
>
> We have conducted experiments that address the specific claims discussed, demonstrating how our algorithm outperforms existing baselines. Below, we outline the issues listed in the introduction and the corresponding experimental evidence provided in the paper:
>
> * Quantifying uncertainty: We measured calibration error across all experiments. This metric, introduced in Appendix 8.6.1, evaluates the reliability of probabilistic predictions.
>
> * Heterogeneity: The PV-EW dataset used in our experiments is a heterogeneous dataset as it comprises clients from two distinct modes. Figure 5 illustrates the claim that heterogeneity is significant in this dataset. Our results show superior performance in such challenging settings compared to state-of-the-art methods.
>
> * Overfitting due to small datasets: We performed experiments on the PV-S, PV-EW, and FEMNIST datasets, each with two configurations: (a) using a large dataset and (b) using a small dataset prone to overfitting. Our results demonstrate that our algorithm effectively mitigates overfitting under the small dataset setting, where standard federated learning methods often fail.
>
> * New data incorporation: Results for new clients are reported across all datasets, illustrating how our algorithm improves generalization for unseen clients.
>
> In the updated version of the paper, we have added the datasets reflecting each claim made in the introduction to Table 1 to further clarify the efficacy of our experiments.
>
> **3. The datasets are quite tiny, how do the techniques scale with larger datasets?**
>
> The submitted paper presents experiments on the EMNIST dataset, which includes 80 clients, each with at least 500 data samples. This dataset is relatively large and was chosen to demonstrate the scalability and effectiveness of our method.

---

> ### Author Response · Authors · 2024-12-06
> **Response to reviewer's comments - Part 2**
>
> **4. There are only 3 datasets on which the approach is evaluated.**
> Our approach was evaluated on eight variants derived from four primary datasets, spanning both regression and classification tasks:
>
> * PV dataset (regression): We evaluate different levels of heterogeneity and different number of samples per client to examine their impact on performance.
>
> * Polynomial dataset (regression): This dataset enables controlled experiments to study the method's behavior under well-understood conditions.
>
> * FEMNIST dataset (classification): We consider different numbers of samples per client to evaluate the framework's robustness under varying client dataset sizes.
>
> * EMNIST dataset (classification): A larger dataset used to evaluate the scalability of the method.
>
> Table 2 at the beginning of the experiments section provides a summary of all datasets and their variants. Section 6 describes the datasets, and Appendix 8.5 offers additional details on their characteristics.
>
> By incorporating both regression and classification tasks and systematically varying the dataset characteristics, such as heterogeneity levels and client data sizes, we believe our experiments demonstrate the versatility of our framework. We hope this clarification addresses your concerns about the diversity of datasets and the comprehensiveness of our evaluation.
>
> **5. Only on one dataset the approach seems to beat meta-learning approaches.**
>
> We highlight that our approach (PAC-PFL) PAC-PFL demonstrates superior performance across the majority of datasets and metrics. To improve the readability of our experimental results, we have merged Tables 3-4 and 7-10 in the previous version of the paper into two new tables in the updated version: Table 3 reporting classification results and Table 6 reporting regression results.
>
> Below is a summary of our results on different datasets:
>
> * Regression datasets: PAC-PFL outperforms all baselines in terms of accuracy across all 5 datasets and demonstrates superior calibration error in 2 out of 5 datasets.
>
> * Classification datasets: PAC-PFL achieves higher accuracy across all 3 datasets and better calibration error in 2 out of 3 datasets.
>
> These results illustrate that PAC-PFL outperforms meta-learning baselines across diverse datasets and metrics, substantiating the efficacy of our approach.
>
> **6. Scalability is explicitly called out when discussing the work by Corinzia & Buhmann (2019). How is PAC-PFL better than that? It should be noted that while the primary concern seems to be "sequential" nature of fine-tuning models at the client, this can effectively be done in parallel, so I am not sure what the authors mean by "scalability".**
>
> We would like to clarify the key distinction in how the two approaches handle client updates, which directly impacts scalability.
>
> The theoretical results proposed by Corinzia & Buhmann (2019) hold when clients update the server model one after the other. Making this algorithm run in parallel by considering synchronous updates of multiple clients is not straightforward and is explicitly mentioned as future work in the Conclusions section of their paper.
>
> In contrast, PAC-PFL is designed to send the model simultaneously to a batch of clients, allowing client updates to be performed in parallel. This parallelization can significantly enhance scalability, enabling our approach to handle a large number of clients efficiently without incurring the bottlenecks associated with sequential processing.
>
> We hope this clarification resolves the concern and highlights the scalability advantage of PAC-PFL over the method by Corinzia & Buhmann (2019).
>
> **7. Define hyper-prior and hyper-posterior formally (perhaps within a definition environment)**
>
> We have added Definition 4.1 for defining the hyper-prior and hyper-posterior distributions in the updated version of the paper. Thank you for your suggestion.

---

> > ### Comment · Reviewer_J4a8 · 2025-01-06
> > **Thank you authors**
> >
> > Thank you authors, for your detailed responses. My comments are sufficiently addressed.

---

### Author Response · Authors · 2024-12-06
**Updated version of the paper**

Dear reviewers,

We would like to thank all reviewers for their valuable feedback on our paper. We have carefully analyzed each question and concern highlighted by the reviewers and provided individual responses to address them.
Based on the received reviews, we have taken significant steps to enhance our manuscript. The updated version of the paper has been uploaded. Thank you for your insightful feedback and continued support. Should you require any further clarification, please do not hesitate to reach out.

---

### Decision · Action_Editor_Ho5Q · 2025-01-16

**Recommendation:** Accept as is

**Comment:**

The work addresses the challenge of Personalized Federated Learning (PFL) in heterogeneous settings. It proposes adapting ideas from PAC-Bayesian meta-learning to develop a novel algorithm for PFL. The proposed approach is specifically designed to handle high client heterogeneity and mitigate overfitting to a central model, mainly when client datasets are small. Additionally, the method is robust to dynamic scenarios, allowing clients to join the training process at later phases rather than all simultaneously.

The reviewers commended the paper for its innovative approach, which bridges two distinct branches of machine learning research. However, they also noted that further investigation is needed to verify whether the claims regarding scalability and extreme levels of heterogeneity hold beyond the ranges studied in this work.

**Audience:**

Yes. The paper introduces a new method for personalized federated learning, a topic within the scope of TMLR's audience.

**Claims And Evidence:**

This paper proposes adapting meta-learning techniques for Personalized Federated Learning (PFL). The properties of the proposed method are thoroughly analyzed through both theoretical analysis and experiments. The claims regarding scalability and heterogeneity are well-supported for datasets with up to 80 clients.